# Diverse Functions of Autophagy in Liver Physiology and Liver Diseases

**DOI:** 10.3390/ijms20020300

**Published:** 2019-01-13

**Authors:** Po-Yuan Ke

**Affiliations:** 1Department of Biochemistry & Molecular Biology and Graduate Institute of Biomedical Sciences, College of Medicine, Chang Gung University, Taoyuan 33302, Taiwan; pyke0324@mail.cgu.edu.tw; Tel.: +886-3-211-8800 (ext. 5115); Fax: +886-3-211-8700; 2Liver Research Center, Chang Gung Memorial Hospital, Taoyuan 33305, Taiwan; 3Division of Allergy, Immunology, and Rheumatology, Chang Gung Memorial Hospital, Taoyuan 33305, Taiwan

**Keywords:** autophagy, selective autophagy, liver, liver disease, hepatitis, steatosis, fibrosis, cirrhosis, hepatocellular carcinoma

## Abstract

Autophagy is a catabolic process by which eukaryotic cells eliminate cytosolic materials through vacuole-mediated sequestration and subsequent delivery to lysosomes for degradation, thus maintaining cellular homeostasis and the integrity of organelles. Autophagy has emerged as playing a critical role in the regulation of liver physiology and the balancing of liver metabolism. Conversely, numerous recent studies have indicated that autophagy may disease-dependently participate in the pathogenesis of liver diseases, such as liver hepatitis, steatosis, fibrosis, cirrhosis, and hepatocellular carcinoma. This review summarizes the current knowledge on the functions of autophagy in hepatic metabolism and the contribution of autophagy to the pathophysiology of liver-related diseases. Moreover, the impacts of autophagy modulation on the amelioration of the development and progression of liver diseases are also discussed.

## 1. Introduction

Autophagy is an evolutionarily conserved process that catabolizes intracellular components through lysosomes to recycle nutrients for supplying energy and regenerating organelles [1,2]. Several types of stress and damage stimuli, such as the deprivation of nutrients, the damage of organelles, the unfolding and aggregation of proteins, and tissue injury have been shown to induce autophagy [3,4]. Interference with the precise and appropriate process of autophagy may contribute to the pathogeneses of various human diseases, such as liver-associated diseases, neurodegenerative diseases, cancer, and infectious diseases [5,6]. In the past few decades, the homeostatic role of autophagy has emerged in the regulation of liver physiology through promoting the degradations of macromolecules and organelles to support the balance of energy as well as the metabolism and regeneration of organelles [7,8,9]. Additionally, autophagy has been indicated as a disease-associated factor that is modulated in the liver cells of people with liver-related diseases, and it contributes to the development and progression of various liver diseases, including hepatitis, steatosis, fibrosis, cirrhosis, and hepatocellular carcinoma [7,10,11,12]. Autophagy protects liver cells against injury and cell death by eliminating the damaged organelles and proteins that are introduced in those with liver-associated diseases. Conversely, autophagy could also act as an alternative pathway that promotes the development and progression of liver diseases. Most importantly, the modulation of autophagy has been extensively proved to alter the occurrence and outcome of liver-related diseases, implying that it represents a novel therapeutic target for the design of new and effective therapies to prevent and treat liver diseases. In this paper, we summarize the current knowledge on the functional role of autophagy in liver physiology and address how autophagy is regulated by liver-associated diseases to become involved in the prevention or promotion of disease occurrence and pathogenesis.

## 2. Overview of Autophagy

The term autophagy is derived from the Greek words for auto (“self”) and phagy (“eating”). The concept of autophagy was initially devised from the observation of vesicle-like dense bodies that encompass cytoplasmic organelles, such as mitochondria and endoplasmic reticulum (ER), in differentiated kidney tissue in mice and glucagon-perfused rat hepatocytes viewed using transmission electron microscopy (TEM) [13,14,15,16]. These double-membraned dense bodies were shown to be associated with the lysosome-mediated degradative process [13,14,15,16]. Subsequently, this process was termed “autophagy” by de Duve, the 1974 Nobel Laureate in Physiology or Medicine, at the Ciba Symposium on Lysosome in 1963 [17,18]. In the late 1960s, several studies, through morphological and biochemical characterization, revealed that glucagon induces the formation of autophagic vacuoles, which are influenced by lysosomes and lysosomal enzymes [19,20]. Despite the effects of glucagon, the deprivation of amino acids and growth factors was indicated to trigger autophagy [21,22,23,24]. From the 1970s to the 1990s, numerous studies demonstrated that the induction of autophagy enhanced the degradation of long-lived proteins, leading to a decrease in amino acid levels [24,25,26]. Additionally, the molecular signaling underlying autophagy initiation and the autophagy inhibitors generated from these findings, such as 3-methyladenine (3-MA) and okadaic acid, have been identified and characterized [24,27,28,29,30,31,32,33]. The membrane source of support for a phagophore for the emergence of autophagic vacuoles was first described in the late 1980s [34] and was further characterized in the 1990s [35,36,37,38,39,40,41]. The comprehensive isolation and molecular cloning of autophagy-related genes (ATGs) were initiated by Ohsumi, who was the 2016 Nobel Laureate in Physiology or Medicine for work on the genetic screening of temperature-sensitive, autophagy-defective mutants in *Saccharomyces cerevisiae* [42,43,44]. Ohsumi identified 15 autophagy-defective mutants that can be respectively complemented by the corresponding ATGs, which function in the entire process of yeast autophagy and degradation [43]. Analogous to yeast, the functional ATGs involved in autophagy in humans and other eukaryotes were also identified and characterized [45,46,47,48,49]. To date, approximately 40 ATGs have been identified [49,50,51], most of which have been evolutionarily conserved among almost all eukaryotes. Furthermore, the nomenclature for ATGs across different species of eukaryotes has been unified [45,46,47,48,49].

### 2.1. Three Modes of Autophagy

Three types of autophagy have been defined according to the mechanism used for the delivery of the intracellular components to lysosomes for degradation: microautophagy, chaperone-mediated autophagy (CMA), and macroautophagy (Figure 1) [1,2]. Microautophagy was defined in mammalian cells through TEM observation of a lysosomal membrane rearranged to have a protrusion and arm-like structure to wrap the cytoplasmic portion into the lumen of the lysosome for decomposition (Figure 1) [17,52,53]. Microautophagy not only randomly engulfs the intracellular materials to instigate degradation (so-called nonselective microautophagy) but also selectively eliminates specific organelles (defined as selective microautophagy) in yeast cells [54,55]. Although core ATG proteins and the endosomal sorting complexes required for transport (ESCRT) machinery are required for microautophagy [56,57,58,59,60], information about how microautophagy is precisely induced and the detailed molecular mechanisms underlying the process of microautophagy remain limited. Similarly, the functional role of microautophagy in human health and diseases is also largely unknown and requires further investigations. CMA is characterized by a selective elimination process in which the degradative substrates that contain the pentapeptide “Lys-Phe-Glu-Arg-Gln” (KFERQ) motifs are specifically recognized by a cytosolic chaperone, namely, the heat-shock cognate protein of 70 kDa (HSC70); these motifs are transported into the lysosomal lumen through the lysosomal membrane protein 2A (LAMP2A)-mediated docking process (Figure 1) [61,62]. Multiple types of stress have been shown to induce CMA, such as nutrient starvation, DNA damage, hypoxia, oxidative stress, and metabolic imbalance [63,64,65,66,67,68]. Crucially, CMA plays a role in the replenishment of amino acids and ATP in cells that have undergone prolonged starvation [64,69], the regulation of lipid metabolism [70,71], the reprogramming of gene transcription [72,73,74], the activation of immune responses [75,76], the control of cell cycle progression [68,77], and the control of ageing [78,79]. Accordingly, the malfunctioning of CMA has emerged as a contributor to numerous human diseases, such as tumorigenesis [80,81,82,83], neurodegenerative disorders [84,85,86,87,88,89], liver diseases [90,91], and lysosomal storage disorders [92]. In macroautophagy (hereafter referred to as autophagy), the membrane rearrangement process leads to the formation of an autophagosome, a double-membranous vacuole that sequestrates the cytoplasmic components and delivers them to lysosomes for degradation (Figure 1) [2,93]. Several types of stress, such as the starvation of nutrients, damage of organelles, aggregation of proteins, and invasion of pathogens, have been shown to induce autophagy [3,4]. In the past decade, autophagy has emerged as a “double-edged sword” in the pathogenesis of a variety of human diseases, including neurodegenerative diseases [94,95,96,97], cancer [98,99], cardiovascular diseases [100,101,102], ageing [94,99,100,101,102,103,104], infectious diseases [105,106], and metabolic disorders [98,107,108,109,110]. Therefore, targeting autophagy could be a feasible strategy for treating human diseases.

### 2.2. Stepwise Process of Vacuole Biogenesis for Autophagy

Autophagy undergoes a stepwise process for vacuole biogenesis that involves the initial nucleation and elongation of the isolation membrane (IM)/phagophore, the closure of autophagosomes, and the fusion of autophagosomes with lysosomes to form autolysosomes (Figure 1) [111,112,113]. Numerous organelles [112,113,114], including the ER [115,116], Golgi apparatus [117], mitochondria [118], recycling endosome [119,120], plasma membrane [121], and mitochondria-associated ER membrane (MAM) [122] support the membrane source for the emergence of the IM/phagophore. At the initial stage, the IM/phagophore that originates from a particular membrane structure, which is often derived from the ER, expands to form a double-membraned and enclosed autophagosome (Figure 1) [114,123,124,125]. Subsequently, mature autophagosomes fuse with the lysosome to generate autolysosomes in which the interior materials are degraded by lysosomal proteases (Figure 1) [124,126,127,128].

Most ATGs (also known as core ATGs) and the signaling molecules and vesicle-trafficking factors involved in other cellular pathways are coordinately required for the completion of the entire autophagic process (Figure 1) [93,129,130]. The starvation of nutrients in cells leads to the suppression of the mammalian target of rapamycin (mTOR), a serine/threonine protein kinase required for controlling cell growth [131,132]. The repression of mTOR results in the translocation of the unc-51 like-kinase (ULK) complex (contains ULK1/2, ATG13, RB1-inducible coiled-coil 1 (RB1CC1, also known as FIP200) and ATG101( from the cytosol to a certain domain reconstituted from the ER (Figure 1) [133,134]. This translocation of the ULK complex in turn recruits the class III phosphatidylinositol-3-OH kinase (PI(3)K) complex (class III-PI(3)K, including Vps34/PI(3)KC3, Vps15, Beclin 1, and ATG14) to the ER membrane-derived domain (Figure 1) [130,135,136] and enhances the activity of the PI(3)K complex through the phosphorylation of Vps34/PI(3)KC3 [137]. PI(3)K in turn produces phosphatidylinositol-3-phosphate (PtdIns(3)P), leading to the recruitment of double-FYVE-containing protein 1 (DFCP1) and WD-repeat domain PtdIns(3)P-interacting (WIPI, the mammalian orthologue of ATG18) family proteins to promote the organization of an ER-associated omegasome structure (also termed IM/phagophore) (Figure 1) [130,135,136,138,139]. Moreover, two multi-spanning membrane proteins, namely ATG9a and vacuole membrane protein 1 (VMP1), are critical to the initial biogenesis of autophagosomes. The ATG9a-enriched vesicles that are trafficked from the trans-Golgi network (TGN) to the ER deliver the lipid bilayers required for autophagosome formation [140,141,142]. ER-associated VMP1 interacts with Beclin 1 of the PI(3)K complex, thereby facilitating the generation of PtdIns(3)P required for the assembly of IM/phagophore [143,144,145]. The subsequent expansion and enclosure of the IM/phagophore into a mature autophagosome requires two ubiquitin-like (UBL) conjugation systems (Figure 1) [146,147,148,149]. The ATG5-ATG12 conjugate is formed by the ATG7 (E1) and ATG10 (E2) enzymatic cascade (Figure 1). This conjugate then forms a trimeric complex with ATG16L (an ATG12-ATG5-ATG16L complex) [146,147,150,151,152]. The other conjugation is that of the phosphatidylethanolamine (PE)-conjugated ATG8 family proteins (including the microtubule-associated protein 1 light chain 3 (LC3) and gamma-aminobutyric acid receptor-associated protein (GABARAP) subfamilies). After protein translation, the C-terminal region of ATG8/LC3 family proteins are immediately processed by ATG4 family proteases to form ATG8/LC3-I. Then, ATG7 enzyme 1 (E1) and ATG3 enzyme 2 (E2) confer the conjugation of the ATG8/LC3-I to generate PE-ATG8/LC3, sometimes called ATG8-LC3-II (also known as lipidated ATG8-LC3) (Figure 1) [153,154,155]. PE-ATG8/LC3 participates in the elongation of the autophagosomal membrane [156] and the tethering and membrane fusion of autophagic vacuoles [149]. Notably, ATG5-ATG12 may act as an E3-like enzyme to promote the lipidation of ATG8/LC3 [157,158], thereby promoting the formation of autophagosomes. Additionally, the sphingolipid microdomains, so-called lipid rafts, were indicated to play roles in the morphogenesis of autophagic vacuoles [159]. The fluorescence resonance energy transfer (FRET) and co-immunoprecipitation (co-IP) studies showed that ganglioside GD3, a paradigmatic raft constituent, interacts with PI3P and LC3-II on the immature autophagosomal membrane [159]. Also, the interactions between GD3 and WIPI/ATG18 family proteins as well as autophagy and Beclin 1 regulator 1 (AMBRA1) were shown in MAM raft-like microdomains [160]. Downregulation of GD3 level by gene knockdown of ST8SIA1/GD3 synthase and alteration of sphingolipid metabolism by fumonisin B1 was demonstrated to inhibit autophagic process [159] and interfere with the interaction of AMBRA1 with calnexin at MAM [160], implying that MAM-associated lipid rafts function in the biogenesis of autophagosomes.

The mature autophagosome fuses with a lysosome, forming an autolysosome in which acidic proteases degrade the sequestrated materials to recycle their nutrients. The autophagosome–lysosome fusion process relies on the multilayered actions of protein–protein interactions, microtubule-mediated transport, and membrane fusion events [123,125,126,128]. The actions of the microtubule ensures the precise transport of the autophagosome to the lysosome for fusion [128,161,162]. The small GTPase Ras-related protein 7 (Rab7) located on the autophagosomal membrane interacts with FYVE and coiled-coil domain-containing 1 (FYCO1) and Rab-interacting lysosomal protein (RILP), two effectors that are respectively linked to kinesin and dynactin in microtubules [163,164,165,166,167], enabling the movements of the autophagosomes on microtubules (Figure 1). Apart from microtubules, the histone deacetylase 6 (HDAC6)-induced remodeling of F-actin and the formation of the F-actin network also promote autophagosome–lysosome fusion in the quality control autophagy-mediated removal of aggregated proteins rather than starvation-induced autophagy [168]. In addition to bridging the transport of autophagosomes on microtubules, Rab7 located on late endosomes and lysosomes stimulates autophagosome–lysosome fusion through recruiting several effectors of this action, including the pleckstrin homology domain-containing protein family member 1 (PLEKHM1) and the homotypic fusion and protein sorting (HOPS) complex (Figure 1) [153,154]. PLEKHM1 contains an LC3-interacting motif that can bind to ATG8 family proteins located on the autophagosomal membrane and concomitantly interacts with Rab7 as well as the HOPS complex, thereby facilitating the fusion of autophagosomes and lysosomes (Figure 1) [169]. Additionally, the PI(3)K complex-associated UV radiation resistance-associated gene (UVRAG) binds to the HOPS complex via Vps16 to induce Rab7 GTPas activity and trigger autophagosome–lysosome fusion [136,170,171]. Notably, the binding of Rubicon to the PI(3)K protein complex reciprocally interferes with the fusion of autophagosomes and lysosomes [136]. Another protein complex, containing ATG14L, syntaxin 17 (STX17), synaptosome-associated protein 29 (SNAP29), and vesicle-associated membrane protein 8 (VAMP8), also stimulates autophagosome–lysosome fusion, presumably through the membrane tethering and fusion process (Figure 1) [172,173]. Recently, ATG8 family proteins were shown to be mainly active during autophagosome–lysosome fusion rather than autophagosome biogenesis at the initial stage of autophagy by recruiting PLEKHM1 in PTEN-induced putative kinase 1 (PINK1)/Parkinson’s disease protein (Parkin)-mediated autophagic clearance of mitochondria (so called mitophagy) and starvation autophagy [174]. In addition to acting at the biogenesis of autophagosomes [159], lipid rafts have emerged as a regulator of autolysosome maturation [159,160]. The interaction between GD3 and lysosome-associated membrane protein-1 (LAMP1) in the autolysosomal membrane as demonstrated by FRET, co-IP, and TEM assays indicated that GD3-enriched lipid rafts could induce membrane remodeling to promote the morphogenesis of autolysosomes and increase autophagic flux [160].

After degradation within autolysosomes, the nutrient-fed-reactivation of mTOR suppresses autophagy initiation and concomitantly initiates autophagic lysosome reformation (ALR), thereby terminating autophagy [175]. Related studies have implied that spinster (spin), a lysosomal efflux permease, is required for ALR formation [176]. Recently, the Cullin 3-Kelch-like protein 20 (KLHL20) ubiquitin ligase was also shown to participate in autophagy termination by promoting the turnover of the ULK1 and Vps34 complexes [177]. Nevertheless, the detailed molecular mechanism underlying the biogenesis of autophagic vacuoles within the entire autophagy process is not comprehensively understood and further investigations are required.

### 2.3. Selective Autophagy and Cargo Recognition

Autophagy has been considered to be a bulky and nonselective degradative process; however, a growing body of literature has indicated that autophagy may selectively sequestrate specific cargos, including organelles and proteins, to induce degradation. This is termed “selective autophagy” [178,179,180]. The concept of selective autophagy was first described in 1973, in a study that showed that a diabetogenic dose of alloxan or streptozotocin induces selective autophagy to degrade β-granules in intermediate cells in the pancreas of rats [181]. At the initial stage of selective autophagy, the specific cargo receptors recognize the degradative cargos that are tagged through polyubiquitination or additional adaptor proteins and then deliver them into the autophagosome through the interaction of cargo receptors with ATG8 family proteins located on the autophagosomal membrane [182,183,184,185]. Numerous cargo receptors of selective autophagy have been identified and characterized, including the neighbor of BRCA1 (NBR1), calcium-binding and coiled-coil domain-containing protein 2 (Calcoco2, also known as NDP52), p62/sequestosome 1 (SQSTM1), and optineurin (OPTN), all of which contain LC3-interacting regions (LIRs) to bind ATG8 family proteins, thus engulfing the cargos into autophagosomes (Figure 2) [184,185,186]. In addition to eliminating degradative substrates through LIR-containing cargo receptors, the potential ATG8-interacting motifs (AIMs) and GABARAP-interacting motifs (GIMs) have been recently found to regulate selective autophagy within ATGs and other cellular proteins [187,188,189,190]. For example, the *Saccharomyces cerevisiae* ATG19 was shown to directly interact with ATG5 through AIMs, and that interaction recruits the ATG5-ATG12-ATG16L trimeric complex, thus enhancing the lipidation of ATG8/LC3 to promote the local biogenesis of autophagosomes to sequestrate the cargos [191].

Regarding the maintenance of the organelle integrity in eukaryotic cells, selective autophagy plays a homeostatic role in the selective elimination of damaged organelles, termed organellophagy [178,180,192], which provides the recycled nutrients for the regeneration of mitochondria, peroxisomes, the ER, lipid droplets (LDs), ribosomes, lysosomes, and nuclei (Figure 2). Numerous stimuli, such as hypoxia [193,194], the accumulation of reactive oxygen species (ROS) [195,196,197], and mitochondrial depolarization [198,199,200], can result in the fission, depolarization, and damage of mitochondria. Mitochondrial damage triggers selective autophagy to degrade the harmful mitochondria in a process known as mitophagy [201,202]. Mitophagy is often initiated without adequate cleavage of PINK1 by presenilin-associated rhomboid-like protein (PARL) within the inner mitochondrial membrane of damaged mitochondria, thus suppressing the degradation of PINK1 [203,204]. This outcome in turn leads to the accumulation of PINK1 on the outer mitochondrial membrane, thereby phosphorylating ubiquitin at serine 65 and then recruiting the ubiquitin enzyme 3 (E3) ligase Parkin [198,199,200,205,206,207]. Subsequently, Parkin ubiquitinates the mitochondrial proteins onto the outer mitochondrial membrane [198,199,200,205,208], thus recruiting specific cargo receptors, such as Calcoco2/NDP52 and OPTN, for the removal of mitochondria through autophagy (Figure 2) [202,209]. The translocation of these cargo receptors also induces the local concentration of phagophore-organization effectors, including DFCP1 and WIPI/ATG18 family proteins, for autophagosome maturation proximal to the damaged mitochondria [209]. Additionally, TANK binding kinase 1 (TBK1) participates in the cargo recognition process of mitophagy by phosphorylating p62/SQSTM1 at serine residue 403 and OPTN at serine residues 177, 473, and 513 [210,211,212]. Despite the PINK1/Parkin-induced ubiquitination of damaged mitochondria, several outer mitochondrial membrane proteins, including FUN14 domain-containing 1 (FUNDC1), BCL2/adenovirus E1B 19 kDa protein-interacting protein 3 (BNIP3), BCL2/adenovirus E1B 19 kDa protein-interacting protein 3-like (BNIP3L), and yeast ATG32, also activate mitophagy in a ubiquitin-independent manner (Figure 2) [213,214,215,216,217]. Recently, numerous studies have identified novel cargo receptors for mitophagy, such as prohibitin 2 (PHB2) and Toll-interacting protein (Tollip) (Figure 2) [218,219]. In contrast, the deubiquitination (DUB) of mitochondrial proteins onto the outer membrane of mitochondria by DUB enzymes USP30 and USP35 antagonizes mitophagy [220,221].

Selective autophagy promotes the turnover of other intracellular organelles. The specific cargo receptors that confer the elimination of these organelles are also identified and characterized. To degrade oxidized and damaged peroxisomes through pexophagy, yeast ATG36 and mammalian NBR1 and p62/SQSTM1 are required to target the degradative peroxisomes to autophagosomes (Figure 2) [222,223,224,225]. Numerous kinases, such as yeast Hrr25 and mammalian ataxia-telangiectasia-mutated (ATM), induce the phosphorylation of these two cargo receptors, thus promoting the delivery of peroxisomes to the autophagosomal membrane [226,227]. The polyubiquitination of several peroxisomal (PEX) membrane proteins, such as PEX5 and the 70-kDa PEX membrane protein (PMP70), facilitates the recognition of damaged peroxisomes by cargo receptors [227,228].

The targeting of a stressed ER to degradation through ER-phagy involves the biological activities of ATG39, ATG11, and ATG40 [229] in yeast cells; the family with sequence similarity 134, member B (FAM134B) (Figure 2); and reticulon family proteins in mammals [230,231]. ATG39 and ATG11 also participate in the selective degradation of yeast nuclei, termed nucleophagy (Figure 2) [229]. The clearance of protein aggregates by selective autophagy is achieved through the p62/SQSTM1- and HDAC6-mediated recognition of Lys63 (K63)-linked poly-ubiquitination of aggregated proteins [168,232,233,234]. Moreover, NBR1 and autophagy-linked FYVE (ALFY) could cooperate with p62/SQSTM1 to degrade protein aggregates through selective autophagy (Figure 2) [235,236,237,238]. The injured lysosomes have recently been reported to be removed by lysophagy, which begins with the recruitment of galectin-3 and LC3 onto lysosomal membranes, which are subsequently recognized by p62/SQSTM1 and delivered to the autophagosome for degradation (Figure 2) [239,240]. Similarly, selective autophagy has emerged as playing a pivotal role in the clearance of ribosomes, termed ribophagy (Figure 2) [241,242], and in the catabolism of LDs for maintaining metabolic homeostasis (Figure 2) [243,244].

In addition to organellophagy, the cargo receptors of selective autophagy can eliminate specific proteins and invading pathogens. The nuclear receptor coactivator 4 (NCOA4) has been recently shown to interact with ATG8 family proteins and to target ferritin heavy and light chains for autophagic degradation, thus modulating the intracellular level of iron (Figure 2) [245,246]. The turnover of ferritin through selective autophagy, termed ferritinophagy, has been implicated in the regulation of erythropoiesis and DNA replication in blood cells [247,248]. The elimination of infectious pathogens by xenophagy represents the host’s first-line defense in restricting microbial infections [249,250,251]. Pexophagy involves the engulfment of invading pathogens by p62/SQSTM1-, Calcoco2/NDP52-, and OPTN-mediated recognition processes and delivery to the autophagosome for degradation (Figure 2) [106,252,253]. The phosphorylations of p62/SQSTM1 (at serine residues 349 and 403) and OPTN (at serine 177) promote the clearance of infecting pathogens through pexophagy [253,254,255,256]. Taken together, selective autophagy not only maintains cellular homeostasis by removing damaged organelles but also acts as a host defensive mechanism to counteract pathogen infection.

### 2.4. Autophagy as an Alternative Cell-Death Pathway

Autophagy (“self-eating”) has been considered a stress-responsive, survival mechanism to protect cells against apoptosis (“self-killing”, type I cell death) [257,258,259]. Autophagy is often activated by the inhibition of apoptosis. For instance, simultaneous gene knockout of BAX and BAK, two BCL2 family proteins involved in cell apoptosis in mice was shown to activate autophagy to counteract etoposide (an inhibitor of topoisomerase-2)-induced cell death [260]. Reciprocally, apoptosis can be activated by inhibiting autophagy. Interference with autophagy by gene silencing and pharmacological inhibitors in nutrient-starved cells was shown to trigger cell apoptosis [261]. The specific gene knockout of ATG5 in neuron cells and T cells in mice was demonstrated to increase apoptotic cell death [262,263]. However, autophagy confers an alternative route to promote cell death, known as type II cell death under some specific cellular conditions [264,265]. For instance, human immunodeficiency virus (HIV) infection leads to autophagy activation to trigger apoptotic cell death of CD4/CXCR4-expressing T cells [266]. The inhibition of HIV Env-induced autophagy by gene knockdown and pharmacological inhibitors was demonstrated to interfere with cell apoptotic death [266]. In spite of apoptosis, autophagy was also indicated to promote necrotic cell death [267]. Autophagy was demonstrated to be activated by caspase inhibition to promote cell death through the accumulation of ROS and degradation of catalases [267]. This caspase inhibition-induced cell death could be reversed by interference with autophagy by siRNAs against ATGs and autophagy inhibitors [267]. Notably, the enhancement of cellular autophagy by the Tat-Beclin1 peptide was specifically demonstrated to trigger the “autosis” cell death pathway, which is mediated by the Na^+^, K^+^-ATPase pump and is characterized by the convolution of nuceli at the early-stage and focal swelling of the perinuclear space at the late-stage [268,269]. Besides the autophagy-inducing Tat-Beclin1 peptide, starvation and in vivo cerebral hypoxia-ischemia were also shown to induce autotic cell death [268,269]. These studies together indicate that autophagy not only adapts to stresses to avoid cell death but also induces diverse types of cell death pathways to kill cells when cells no longer circumvent certain stimuli.

## 3. Regulation and Functional Roles of Autophagy in Liver Physiology

### 3.1. The Leading Discovery of Autophagy in Liver Tissue

Hepatocytes in liver tissue were initially revealed to contain autophagic vacuoles. In the early 1960s, Ashford et al. first demonstrated that glucagon perfusion in rats can induce the formation of polymorphic dense bodies in liver cells (Table 1) [14]. These dense bodies were shown to sequestrate the fragmented and morphologically abnormal mitochondria, which were associated with autolysis triggered by the glucagon-related protein catabolic process (Table 1) [14]. Similarly, the treatment of rat livers with the detergent Triton also led to the formation of dense bodies (known as cytolysomes) that exhibit two patterns: one consists of double-membraned vacuoles containing mitochondria and ER membrane fragments and the other consists of single-membrane vesicles in which the engulfed materials are degraded (Table 1) [15]. Soon thereafter, glucagon was revealed to be as an activator of autophagy in liver cells (Table 1) [19,20,270], which were standardized for monitoring autophagy. De Duve and Deter first observed that glucagon administration triggers an increase in lysosomal size, which could be related to the formation of autophagic vacuoles in the rat livers (Table 1) [19]. After the biochemical fractionation of lysosomes, their study further revealed that glucagon induction upregulates acidic phosphatase as well as cathepsin D in lysosomes and also increases the fragility of lysosomes in Rat liver (Table 1) [19]. In a subsequent study that combined biochemical fractionation and TEM, Deter and colleagues revealed by a morphological quantification that glucagon-induced autophagic vacuoles represent a substantial portion of lysosomes in liver homogenates (Table 1) [20]. Their study implied that hepatic lysosomes are involved in the biogenesis of autophagic vacuoles and thus provide the main source of acidic proteases for the degradation of sequestrated interior materials (Table 1) [20]. Moreover, two types of glucagon-triggered autophagic vacuoles in the liver were further specified: type I vacuoles are predominantly double-membraned vacuoles that contain the ER, ribosomes, and ground cytoplasm and type II vacuoles are larger than type I vacuoles and are composed of a single limiting membrane, in which the sequestrated ER and cytoplasm are broken down (Table 1) [270]. Taken together, these studies not only indicate that hepatic autophagy may present a novel degradative process that eliminates the intracellular components in the liver but also provide evidence that lysosomes participate in autophagy to support proteolytic enzymes.

### 3.2. The Role of Autophagy in Balancing Metabolism and Sensing Stresses in the Liver

The regulation of autophagy in liver physiology and the modulation of autophagy by liver injury were discovered in the early 1970s [271,272,273,274]. The study by Pfeifer first revealed the role of autophagy in the decomposition of glycogen in liver atrophy [273]. Long-term starvation has been shown to induce hepatic autophagy, correlating with cell atrophy in rat livers (Table 1) [274,275]. These studies suggested that hepatic autophagy may detect malnutrition in the liver as well as liver damage, instantly supporting the refueling of nutrients through degradation. Accordingly, the formation of autophagic vacuoles was shown to be energy-dependent and correlated with the rate of protein synthesis (Table 1) [276,277], alteration of metabolites (Table 1) [278,279,280], and interference with cytoskeleton organization (Table 1) [281]. In the late 1970s, stress and amino acid deprivation were demonstrated to trigger autophagy in hepatic cells (Table 1) [21,25,280]. This autophagic proteolytic effect induced by the deprivation of nutrients in hepatocytes can be inhibited by the refeeding of nutrients and autophagy inhibitors (Table 1) [24,30,282], indicating that the status of nutrient supplies plays a detrimental role in autophagy activation in the liver. These studies collectively imply that autophagy acts as a regulator that senses changes in the metabolism and alterations of energy in the liver.

The induction of hepatocellular necrosis by dimethylnitrosamine (DMNA) can increase the number and size of autophagic vacuoles in the period beyond the onset of cell necrosis (Table 1) [271,272], suggesting that autophagy might be activated to counteract cell death in the liver. At the same time, numerous studies have shown that the smooth membrane of the ER can contribute to the membranous structure that supports autophagosome biogenesis in liver cells [283,284,285,286,287], leading to a new paradigm for understanding the membrane resource for developing autophagosomal membranes. Collectively, these studies indicate that hepatic autophagy could be activated by numerous stimuli, such as nutrient starvation, metabolism imbalance, and liver injury, to promote the maintenance of metabolic homeostasis.

### 3.3. Turnover of Macromolecules through Autophagy in the Liver

In the late 1970s, autophagy was first shown to degrade glycogen and to participate in the selective elimination of organelles in the liver (Table 1) [23]. In line with this study, biochemical and morphological studies have, together, demonstrated that hepatic autophagy plays a major role in protein degradation and the degeneration of organelles through the formation of autolysosomes (Table 1) [23,25,287,288,289,290,291,292,293,294]. The functional roles of autophagic degradation in the liver were implicated in the turnover track of intracellular macromolecules, such as the degradation of fetal-type glycogen in the neonatal period (Table 1) [295], the destruction of damaged organelles by virus infection (Table 1) [325], the selective degradation of RNA and proteins through the deprivation of amino acids (Table 1) [296], and the elimination of the ubiquitin–proteasomal pathway through long-term starvation (Table 1) [297]. Additionally, hepatic autophagy is involved in multiple cell surveillance mechanisms, including the regulation of ischemic liver injury (Table 1) [298], the growth suppression of carcinogen-treated hepatocytes (Table 1) [299], the modulation of the iron pool and sensitivity to oxidative stress (Table 1) [300], and the regulation of cell death in the damaged livers of patients with anorexia nervosa (Table 1) [301]. Conversely, autophagy plays critical roles in the integration of metabolic pathways by regulating the supply of amino acids for effective translation in hepatoma cells (Table 1) [302], the balancing of blood glucose and amino acid levels (Table 1) [303], and the activation of hepatic stellate cells (Table 1) [304]. Moreover, autophagy participates in the regulation of lysosomal proteolysis in liver regeneration (Table 1) [305], the degeneration of transplanted livers in rats (Table 1) [306], the maintenance of hepatic function in the aged liver (Table 1) [79], and the suppression of age-dependent ischemia in injured livers (Table 1) [307]. Taken together, these results indicate that autophagy acts as a protector in physiologically balancing liver metabolism and maintaining liver function and growth.

### 3.4. Selective Degradation of Organelles through Autophagy in the Liver

In the past few decades, numerous studies have indicated that autophagy participates in the catabolism of intracellular compartments in the liver, including Mallory–Denk bodies (MDBs) [308,309,326], LDs [70,71,243,244,310,311,327,328], peroxisomes [312,313,314,315,316,317,318,329,330], mitochondria [320,321,322,323], and the ER [324,331] (Table 1). A biochemical fractionation study indicated that a considerable portion of several types of organelles was sequestrated within autophagic vacuoles in rat hepatocytes (Table 1) [332,333], implying the functional roles of autophagy in the elimination of intracellular organelles in the liver. MDBs are cytosolic hyaline inclusions that were discovered in the hepatocytes of patients with alcoholic hepatitis in 1911 by Mallory [334] and further characterized in mouse livers by Denk in the late 1970s [335,336]. Several intracellular components are enclosed in MDBs, including keratins, chaperones, protein degradation machinery that contains ubiquitin and p62/SQSTM1, and phosphoproteins [337]. MDBs have been observed in various liver diseases, such as alcoholic steatohepatitis, nonalcoholic steatohepatitis (NASH), nonalcoholic fatty liver disease (NAFLD), and hepatocellular carcinoma (HCC) [337,338,339]. Harada et al. first demonstrated that rapamycin-induced autophagy may mediate the turnover of bortezomib-induced MDBs in in vitro cell cultures and in vivo mouse models [308,309], supporting autophagy′s role in the clearance of cytoplasmic inclusions.

LDs are the primary organelles that store neutral lipids, including cholesterol ester and triglycerides (TG), and serve as a reservoir for energy, particularly for the liver [340,341,342]. The aberrant accumulation of lipids in LDs has been evinced in numerous metabolic disorders in the liver, such as hepatic steatosis, NASH, and NAFLD, leading to global health burdens in modern society (Table 1) [328,340,342,343]. The role of autophagy in LD dynamics was originally defined in the analysis by Fujimoto et al. of apolipoprotein B (ApoB) degradation (Table 1) [311]. By combining biochemical fractionation and microscope-based approaches, the authors posited that autophagy may promote the degradation of ApoB, which specifically occurs around the surface of LDs in hepatocytes (Table 1) [311]. Subsequently, Singh et al. showed that interference with autophagy by the knockdown of the ATG5 gene expression increased TG accumulation and inhibited the β-oxidation of free fatty acids (FFAs) and degradation of TG in hepatocytes (Table 1) [244]. Their TEM-based ultrastructural study further indicated that LDs are delivered into autophagic vacuoles for degradation, which is enhanced by nutrient starvation (Table 1) [244]. Their study first uncovered the role of autophagy in the catabolism of LDs, (e.g., “lipophagy”). Another study further confirmed that starvation upregulated lysosomal lipase activity in the autophagic fraction of the liver to promote lipid degradation (Table 1) [310]. In contrast, ATG7 deficiency in mouse hepatocytes was shown to impede the formation of LDs (Table 1) [243]. The specific localization of lipidated-LC3 onto the surface of LDs in starved mouse hepatocytes suggested that the ATG8/LC3-lipidation process might be involved in the biogenesis of hepatic LDs (Table 1) [243]. In line with this study, another report demonstrated that mammalian ATG2 plays a crucial role in the morphogenesis and dynamics of LDs (Table 1) [344]. In addition, autophagy was shown to inhibit ethanol-induced steatosis in mouse livers and to protect liver cells from ethanol-triggered hepatotoxicity (Table 1) [345]. In addition to macroautophagy, CMA was recently indicated to promote the degradation of LD-associated proteins perilipin 2 (PLIN2) and perilipin 3 (PLIN3) to control LDs biogenesis [70]. Moreover, the 5′-AMP-activated protein kinase (AMPK)-induced phosphorylation of PLIN2 was shown to promote its interaction with HSC70, a chaperone of CMA, and thus facilitate the degradation of PLIN2, thereby recruiting lysosomes and cytosolic lipases to catabolize LDs [71]. More importantly, this specific form of LD degradation through autophagy, the so-called lipophagy, was reported to participate in thyroid hormone-induced LD catabolism (Table 1) [346,347].

The sequestration of peroxisomes within autophagic vacuoles was initially observed in a study showing that antilipolytic agent-treated rat livers that contained enhanced autophagic vacuoles engulfed peroxisomes and downregulated the activities of peroxisomal enzymes (Table 1) [312,315], suggesting that autophagy participates in peroxisome degradation. Analogously, autophagic vacuoles have also been reported to sequestrate peroxisomes in the hepatocytes of patients with chronic hepatitis B virus (HBV) who received transplantation and immunosuppressive therapy (Table 1) [329]. Subsequently, amino acid deprivation-induced autophagy was suggested to selectively degrade peroxisomes in hepatocytes isolated from clofibrate-treated rats, and this degradation was completely inhibited by the administration of 3-MA (Table 1) [316]. Furthermore, several other studies have demonstrated that autophagy is involved in the elimination of excess peroxisomes, thus prohibiting the uncontrolled proliferation of peroxisomes in the liver [36,38,313,314,317,330]. Moreover, the selective degradation of peroxisome autophagy, termed “pexophagy”, may regulate peroxisome proliferator-activated receptor α (PPARα) target genes and the β-oxidation of FFAs to prevent hepatic steatosis and tumorigenesis in the liver (Table 1) [348] and acute liver failure induced by inflammation (Table 1) [349].

Mitochondria and the ER are the intracellular organelles that were originally detected in autophagic vacuoles in the late 1950s and early 1960s [13,14,15,16]. The concepts underpinning the degradation of mitochondria through autophagy was derived from the observation that the rate of mitochondria removal was selectively and positively correlated with the formation of autophagic vacuoles in rat livers [23,287]. Subsequently, studies have shown that autophagic vacuoles contain the mitochondrial enzymes of the liver [37,38,350,351,352], further indicating that hepatic autophagy selectively eliminates mitochondria. The autophagic degradation of mitochondria in hepatocytes was further demonstrated to introduce mitochondrial injury, thus promoting the pathogenesis of alpha (1)-antitrypsin (α1-AT) deficiency-induced liver injury that was highly associated with chronic liver diseases (Table 1) [353,354,355]. Komatsu et al. first demonstrated that the genetic knockout of ATG7 in mice interfered with autophagosome biogenesis in livers in which deformed mitochondria had accumulated (Table 1) [356], suggesting that hepatic autophagy plays a major role in mitochondria degradation. The targeting of mitochondria to autophagic degradation was enhanced in the livers of aged mice (Table 1) [321]. Moreover, the impairment of hepatic autophagy was involved in mitochondrial dysfunction in ischemia/reperfusion (I/R)-triggered mouse liver injuries (Table 1) [357]. By contrast, the autophagy-mediated degradation of mitochondria was related to acute liver cell damage in patients with anorexia nervosa (Table 1) [301]. The selective degradation of autophagy, termed “mitophagy”, was demonstrated to underlie mitochondrial remodeling in rat hepatocytes [323]. Mitophagy reduces ethanol-induced toxicity in mouse livers (Table 1) [345], regulates interferon (IFN)-mediated antiviral responses (Table 1) [358], protects liver cells against acetaminophen-induced hepatotoxicity (Table 1) [359], rescues liver function in efavirenz-induced mitochondrial dysfunction (Table 1) [360], suppresses the development of HCC (Table 1) [361], and prevents liver damage in patients with NAFLD (Table 1) [362,363]. Overall, autophagy not only plays a crucial role in the balance of diverse metabolic pathways but also promotes the elimination of damaged organelles and protects liver cells from injury, thereby maintaining cellular homeostasis.

## 4. Autophagy: A Friend or Foe in Liver Diseases?

### 4.1. Liver Injury

The correlation of autophagy and liver injury was first described in studies showing that DMNA-induced liver damage activates the formation of autophagic vacuoles (Table 2) [272]. Subsequently, the formation of autophagic vacuoles was detected in the injured liver cells of mice treated with lysine acetylsalicylate (Table 2) [279], in the livers of mice exposed to acute stressors (Table 2) [364], and in the injured hepatocytes of rats infected with lethal *Escherichia coli* (Table 2) [365]. Numerous physiological and pathological stimuli in the Rat liver were implicated in the elevated autophagy-mediated protein degradation (Table 2) [366]. Alpha (1)-antitrypsin deficiency has been considered as a major cause of liver injury in patients with chronic hepatitis and HCC (Table 2) [367,368,369]. Autophagy has been extensively demonstrated to be activated in the injured liver of α1-AT-deficient mice and may participate in the disposal of mutant α1-ATZ aggregated proteins (Table 2) [353,354,370]. The role of autophagy in the clearance of α1-ATZ mutated proteins was further proven by genetic studies showing that the gene knockout of ATG5 in mice leads to an increased abundance of insoluble α1-ATZ [371] and that the deficiency of ATG6 and ATG14 inhibits α1-ATZ degradation in yeast cells (Table 2) [372]. Conversely, the induction of autophagy by rapamycin may reduce intrahepatic α1-ATZ aggregation and related liver injury in mice (Table 2) [373]. These studies collectively indicate that autophagic degradation plays a pivotal role in the elimination of α1-ATZ aggregates in the cytoplasm to prevent these aggregated proteins from impairing the ubiquitin–proteasomal pathway and to protect liver cells from organelle damage and cell death (Table 2) [355,374,375,376].

Evidence for the physiological significance of autophagy in the clearance of the cytoplasmic inclusion body was uncovered by the study of Komatsu et al. on ATG gene knockout in mice experiments (Table 2) [356,377]. The genetic deletion of ATG7 in mice resulted in the accumulations of ubiquitin- and p62/SQSTM1-containing protein aggregates and abnormal mitochondria in liver cells and caused liver injury (Table 2) [356], implying that autophagy protects liver cells from damage by promoting the clearance of aggregate-prone proteins. In addition, the gene knockout of p62/SQSTM1 in mice livers represses the accumulation of aggregated proteins in such livers and attenuates liver injury, indicating that autophagy prohibits damage of the liver through the p62/SQSTM1-mediated disposal of cytoplasmic inclusion proteins (Table 2) [377]. Moreover, the accumulated p62/SQSTM1 through autophagy deficiency was shown to interact with Kelch-like ECH-associated protein 1 (Keap1) and interfere with the Cullin3-Kepa1 ubiquitin E3 ligase-mediated proteasomal degradation of nuclear factor erythroid 2-related factor 2 (Nrf2), thereby stabilizing and translocating Nrf2 into the nucleus to transcriptionally activate antioxidant genes expressions (Table 2) [378]. Liver dysfunction in autophagy-deficient mice was further exacerbated by an additional knockout of Keap1 (Table 2) [378]. The upregulation of the p62/SQSTM1-containing aggregate and induction of Nrf2-targeted genes were detected in a major group of HCC cell lines (Table 2) [379]. The induction of liver injury through autophagy deficiency may be associated with the upregulation of oxidation stress, as demonstrated by the high levels of oxidative stress-inducible proteins detected in mouse livers lacking the ATG7 gene expression (Table 2) [380]. Moreover, a reduction of oxidative damage by hepatic autophagy represses ischemic liver injury (Table 2) [381]. These aforementioned studies indicate that autophagic degradation in the liver eliminates aggregate-prone proteins to prevent liver injury. Furthermore, the deregulation of autophagy may induce liver damage and progressive liver diseases.

However, autophagy was shown to be activated in chemotherapy-injured livers to limit the necrotic cell death of hepatocytes (Table 2) [382,383]. Moreover, it was implicated in the repression of age-dependent ischemia and reperfusion-induced liver injury in mice (Table 2) [307]. Autophagy was also suggested to reduce acute ethanol-induced hepatotoxicity in mouse livers by promoting damage to mitochondria through mitophagy (Table 2) [345,384]. Sepsis and lipopolysaccharide (LPS)-induced autophagy via heme oxygenase-1 (HO-1) signaling also protects hepatocytes from death (Table 2) [385]. Moreover, autophagy is also involved in the inhibition of lipotoxicity in the hepatocytes of injured livers (Table 2) [386], in the enhancement of cytotoxicity in polyethyleneimine (PEI)-triggered liver damage (Table 2) [387], and in the protection of livers against acetaminophen (APAP)-induced hepatotoxicity (Table 2) [359,388]. These results collectively suggest a protective role of autophagy in the suppression of liver injury caused by various stimuli. By contrast, other studies have shown the opposite effect of autophagy in liver dysfunction, such as its contribution to cell death during liver graft dysfunction (Table 2) [389] and its involvement in liver cell death in patients with anorexia nervosa (Table 2) [268,301].

Numerous signaling pathways have been shown to activate autophagy during liver injury, such as insulin-like growth factor-1 (IGF-1) signaling [390]; gene transfer of transcription factor EB (TFEB) activity [391]; caspase 1 activation [392]; activation of NACHT, LRR, and PYD domains-containing protein 3 (NLRP3) inflammasome [393]; suppression of protein kinase C (PKC) downstream signaling [394]; and ER stress [395,396,397] (Table 2). However, the NAD-dependent deacetylase sirtuin-1 (Sirt1)-dependent downregulation of circulating high mobility group protein B1 (HMGB1) [398,399], activation of PPARα [349], nuclear receptor binding factor 2 (NRBF2)-mediated activation of the PI(3)K complex [400], AMPK activation [401,402,403], hypoxia-inducing factor-1α (HIF-1α) [404], retinoic acid receptor α (RARα) [405], HO-1 signaling [406,407], suppression of c-jun-N-terminal kinase (JNK) [408], nicotinic acid adenine dinucleotide phosphate (NAADP)-mediated calcium signaling [409], and Krüppel-like factor 6 (KLF6)-mediated transcription [410] were shown to participate in the autophagy-mediated protection against liver injury (Table 2). Nevertheless, these studies collectively indicate that autophagy plays a critical role in protecting liver cells against different types of liver injury. Furthermore, they demonstrate that autophagy represents a potential target for the development of new therapeutic agents for treating liver diseases.

### 4.2. Steatosis and Fatty Liver Diseases

Autophagy promotes LD catabolism (Table 3) [70,71,244,310,311,346,347], and the components of the autophagic machinery were shown to participate in the biogenesis of LDs (Table 3) [243,344]. These studies, thus, imply that hepatic autophagy has a homeostatic role in the regulation of lipid metabolism to prevent liver steatosis. Moreover, it is also a therapeutic target for developing novel therapies for curing fatty liver diseases [11,12,328,411]. An earlier report showed that interference with autophagic degradation in the fatty livers of rats was correlated with tissue necrosis and the limitation of mitochondrial injury (Table 3) [412]. Ding et al. provided the first evidence that the repression of autophagy in the liver by pharmacological inhibitors and RNA interference resulted in the accumulation of LDs and induced apoptosis of hepatocytes of mice (Table 3) [345], suggesting that autophagy attenuates the formation of alcoholic fatty liver. In line with this finding, another study demonstrated that the cytochrome P450 2E1 (CYP2E1) mediated the upregulation of oxidative stress-suppressed autophagy, thus leading to lipid accumulation in cultured liver cells (Table 3) [413]. The inhibition of autophagy by a thymidine analog was shown to lead to lipid accumulation, increased ROS, and hepatic dysfunction (Table 3) [414]. Autophagy was also demonstrated to be activated by exendin-4 to combat dysfunctional ER stress and lipid accumulation in unsaturated fatty acid-induced NAFLD mice (Table 3) [386,415], indicating that autophagy prevents steatosis under NAFLD conditions. The ablation of starvation-induced autophagy by the liver-specific deletion of Vps34, the kinase of the PI(3)K complex in mice, led to the development of hepatic steatosis and hepatomegaly (Table 3) [416]. Additionally, the tumor suppressing p73-mediated transcription of ATG5 promotes autophagy activation to regulate lipid metabolism in hepatocytes (Table 3) [417]. The deletion of the acyl-CoA-dependent lysocardiolipin acyltransferase (ALCAT1), an enzyme required for mitochondrial bioenergetics, was proved to promote autophagosome formation and to prevent NAFLD and related metabolic disorders in mice (Table 3) [418]. Transcriptional factor 3 (TFE3) induced lipophagy to alleviate hepatic steatosis (Table 3) [419]. The sterol regulatory element-binding proteins (SREBPs)-patatin-like phospholipase domain-containing enzyme 8 (PNPLA8) axis was demonstrated to activate autophagy to decrease hepatic steatosis in mice with NAFLD (Table 3) [420]. Interference with the ApoB synthesis has been indicated to induce ER stress to trigger autophagy, thus preventing hepatic steatosis in mice (Table 3) [421]. Rubicon was shown to repress the entire autophagic process to induce lipid accumulation and to trigger cell apoptosis in mice with NAFLD (Table 3) [422]. Saturated fatty acids-induced sirtuin 3 (SIRT3) impaired autophagy to contribute to lipotoxicity in hepatocytes (Table 3) [423]. Recently, the suppression of transcription factor EB (TFEB)-mediated lysosome biogenesis and autophagy was demonstrated to promote chronic ethanol uptake-induced hepatic steatosis and liver injury (Table 3) [424].

The enhancement of autophagy using pharmacological approaches has been shown to alleviate liver steatosis and injury in alcoholic and nonalcoholic fatty livers in mice (Table 3) [425,426]. The gluscose-6-phosphatase (G6Pase) deficiency-induced von Gieke’s disease (GSDIa), a common glycogen storage disorder, has been shown to impair autophagy, leading to hepatic steatosis in those with NAFLD (Table 3) [427]. Furthermore, the pharmacological elevation of autophagy indicated that it could reverse lipid accumulation and liver damage (Table 3) [427]. Trehalose, a naturally occurring disaccharide, was shown to inhibit solute carrier 2A to activate autophagy, thus preventing hepatic steatosis (Table 3) [428]. The activation of autophagy by rapamycin was shown to attenuate ethanol-LPS-induced hepatic steatosis and injury (Table 3) [429]. Taken together, these studies evince that the induction of autophagy through pharmacological approaches represents a feasible therapeutic approach for treating fatty liver diseases. In spite of ethanol- and NAFLD-induced hepatic steatosis, autophagy is known to protect hepatocytes from hepatitis C virus (HCV)-induced liver steatosis in patients with chronic HCV (Table 3) [430]. In addition to autophagy, the specific deletion of CMA in mice leads to hepatic glycogen and induced liver steatosis, implying the role of age-dependent CMA in balancing liver metabolism by promoting lipid catabolism [69]. Parkin-mediated mitophagy was reported to protect against alcohol-induced mitochondrial dysfunction, hepatic steatosis, and liver injury in mice (Table 3) [431]. However, a recent study demonstrated that the p62/SQSTM1-mediated recruitment of Cullin-Kepa1-Rbx ubiquitin E3 ligase ubiquitinates mitochondria and induces mitophagy to alleviate liver injury in mice with NAFLD (Table 3) [362]. The liver-specific deletion of FIP200/RB1CC1 was reported to suppress lipid accumulation, to decrease lipogenic gene expression, and to exacerbate LPS and endotoxin-induced liver injury in an NAFLD mouse model (Table 3) [432].

In contrast to the role of autophagy in attenuating liver steatosis and injury, a series of studies has shown that autophagy also promotes the pathogenesis of steatohepatitis diseases. Autophagy in Kupffer cells was shown to be suppressed by hepatic steatosis, and that enhances an inflammatory response to endotoxins (Table 3) [433]. Additionally, chronic alcohol intake was reported to suppress hepatic autophagy and to promote liver steatosis as well as inflammation, which can be reversed by mitochondrial aldehyde dehydrogenase (ALDH2), a detoxification enzyme of ethanol metabolite acetaldehyde (Table 3) [434]. However, in a recent study, mitophagy and ER-phagy independently participated in the progression of NAFLD (Table 3) [363]. In addition, obesity-induced hepatic steatosis inhibited autophagic proteolysis by interfering with the fusion of autophagosomes with lysosomes (Table 3) [435]. Moreover, hepatic steatosis in the livers of patients with NAFLD and in a murine model of NAFLD were shown to impair autophagic flux, which is associated with elevated ER stress and cell apoptosis (Table 3) [436]. Another study reported that the suppressed expressions of cathepsin family enzymes increased p62/SQSTM1 level in patients with NAFLD with autophagic dysfunction and hepatic inflammasome (Table 3) [437]. These results suggest that hepatic steatosis could interfere with the autophagic process to promote disease progression in patients with fatty liver diseases.

### 4.3. Liver Cancer

The relevance of autophagy in the tumorigenesis of liver cancer was first suggested in an ultrastructural microscopy study published in the late 1970s [438]. By using TEM to investigate changes in the subcellular organelles of liver tissues in the different disease progression stages of patients with liver cancer, Hruban’s study revealed that an increased formation of autophagic vacuoles was associated with early-stage carcinogenesis (Table 4) [438]. Subsequently, another study demonstrated that the induction of carcinogenesis in rat hepatocytes mitigated amino acid deprivation-induced autophagic responsiveness and concomitantly allowed carcinogen-treated cells to survive for a longer period, suggesting that autophagy impaired by carcinogens might benefit the cell growth of carcinogen-altered hepatocytes to promote tumorigenesis (Table 4) [439]. Also, autophagic degradation of cytoplasmic constituents was shown to be negatively regulated by the growth of rat hepatoma cells (Table 4) [440,441,442], implying that downregulated autophagy may selectively promote cell growth from normal cells to transformed cancer cells (Table 4) [443].

Moreover, the deregulation of autophagy was reported to be synergistically associated with altered apoptotic activity, tumor malignancy, and poor prognosis in HCC (Table 4) [444,445]. The activation of autophagy by the knockdown of HIF-2α was demonstrated to attenuate cell apoptosis to promote the cell survival of hepatocellular tumor spheroids (Table 4) [446]. The dysregulation of autophagy through the downregulation of the Beclin 1 expression was reported to be involved in HAb18G/CD147, a transmembrane glycoprotein-induced biomarker of tumorigenesis in human hepatoma cells (Table 4) [447]. Notably, autophagy was shown to suppress hepatocarcinogenesis at the dysplastic stage but promoted tumor growth in the tumor-forming stage (Table 4) [448]. However, in one study, autophagy was reported to maintain mitochondrial integrity and to protect cells against oxidative stress in the early stages of tumor development to prevent hepatocarcinogenesis but was reported to downregulate tumor suppressors to promote the development of HCC (Table 4) [449]. Autophagy was activated by the long noncoding RNAs (lncRNAs)/has-miR-30b-5p axis to promote hepatocarcinogenesis (Table 4) [450]. Autophagy was also reported to be involved in the promotion of HCC cell invasion by regulating the epithelial–mesenchymal transition (EMT) (Table 4) [451,452]. Furthermore, it supports intracellular ATP in the cell proliferation of liver cancer cells through activating mitochondrial β-oxidation (Table 4) [453,454].

The induction of autophagy by concanavalin A (ConA) was demonstrated to inhibit tumor nodule formation in mouse livers and to prolong survival (Table 4) [455,456,457], indicating that the modulation of autophagy is a potential target for designing anticancer therapies for liver cancer. Similarly, the induction of autophagy by different chemotherapy drugs, naturally isolated compounds, and photodynamic therapy has been suggested to inhibit hepatoma cell growth and induce cell death (Table 4) [383,458,459,460,461,462,463,464,465,466,467]. Furthermore, the activation of autophagy was reported to mediate the transforming growth factor-β (TGF-β)-induced growth inhibition and cellular apoptosis of HCC [468], the microtubule-associated protein 1S (MAP1S)-mediated suppression of hepatocarcinogenesis [469], the HDAC6-associated induction of liver cancer cell death [470], the inactivated HDAC1-induced cell death of liver cancer cells [471], the oroxylin A-triggered HCC cell death [472], the MLN4924-induced repression of liver cancer cell growth [473], and the repressed regulator of cullins 1 (ROC1)-mediated inhibition of liver cancer cell growth [474] (Table 4). In addition, autophagy was demonstrated to be induced by Hedgehog inhibition to reduce the liver cancer cell growth induced by several stimuli (Table 4) [475].

Numerous studies have also suggested that autophagy facilitates cell death in human HCC cells treated with a group of anticancer molecules (Table 4) [476,477,478,479,480,481,482,483,484,485]. By contrast, interference with hepatic autophagy can enhance antitumor drug- and biologically active compound-triggered deaths of HCC cells (Table 4) [486,487,488,489,490,491,492,493,494], suggesting that the inhibition of autophagy could be combined with chemotherapy drugs to instigate the death of liver cancer cells. Furthermore, the inhibition of autophagy reportedly represses the development of hepatoblastoma and HCC [495,496], reduces the viability of HCC under hypoxia [497], and induces cell apoptosis in HCC cells by a proteasome inhibitor [498] (Table 4). Autophagy may also contribute to the chemoresistance of HCC (Table 4) [499,500,501,502,503,504,505,506,507]. The suppression of autophagy was found to enhance the susceptibility of liver cancer cells toward sorafenib-induced cell death (Table 4) [508]. Moreover, autophagy was activated by sorafenib and premexetred-combined chemotherapy to enhance the killing of liver cancer cells though also inducing intrinsic cell apoptosis [509]. By contrast, autophagy was reciprocally shown to suppress the chemotherapy-induced apoptosis of HCC cells (Table 4) [510]. Nevertheless, these studies collectively highlight that the modulation of autophagy can be feasibly adapted to form a new therapeutic strategy to cure liver cancers.

The suppressive role of autophagy in the tumorigenesis of liver cancer in in vivo animal models was demonstrated by the studies of Komatsu et al. and Mizushima et al. (Table 4) [379,511]. Komatsu et al. revealed that the liver-specific knockout of ATG7 in mice leads to the development of multiple tumors in the liver, which is accompanied by the accumulation of p62/SQSTM1-induced Nrf2 and downstream transcriptional activation of antioxidant genes (Table 4) [379]. The upregulated p62/SQSTM1-induced Nrf2 activation through Keap1 degradation by autophagy deficiency was analogously detected in the liver tissues of human hepatocellular carcinoma (Table 4) [379]. Similarly, the mosaic deletion of ATG5 in mice was shown to cause multiple liver tumors in which ubiquitin and p62/SQSTM1 were accumulated. Furthermore, oxidative stress and genomic damage responses were elevated (Table 4) [511]. The progression of liver cancer in mice with partial ATG5 deletions was suppressed by the simultaneous knockout of p62/SQSTM1 (Table 4) [511]. Taken together, these results indicate that autophagy is required for the repression of tumorigenesis in liver cancers. A recent study further demonstrated that in the tumor regions of liver tissues in HCV-positive patients with HCC, a high phosphorylation level of p62/SQSTM1 at serine 349 promotes glucoronate pathways and glutathione synthesis through Nrf2-downstream transcription (Table 4) [255,512]. This phospho-p62/SQSTM1-Nrf2 axis-dependent metabolic reprogramming is involved in the survival and chemoresistance of HCC cells and is, thus, a therapeutic target for designing anticancer drugs with improved efficacy (Table 4) [512]. Analogously, p62/SQSTM1 was also induced by diethylnitrosamine (DEN) to enhance carcinogenic activity through Nrf2-induced antioxidant responses and mTORC1 signaling (Table 4) [513]. The accumulated p62/SQSTM1-positive aggregate in the liver tissues contributed to disease progression in NASH and HCC and was correlated with the recurrence of HCC after curative ablation (Table 4) [513]. These results indicate that p62/SQSTM1 could serve as a prognostic biomarker for HCC recurrence after curative ablation (Table 4) [513,514]. Recently, the hepatocyte-specific ablation of ATG7 in mice was shown to promote liver size, fibrosis, the expansion of progenitor cells, and hepatocarcinogenesis, as accompanied by the accumulation of the yes-associated protein (YAP), an effector of the Hippo signaling pathway (Table 4) [515], suggesting that autophagy acts to prevent the progression of liver-related diseases by targeting YAP to induce degradation.

Recently, mitochondrial fission was found to be frequently upregulated in the liver tissues of patient with HCC and related to poor prognosis (Table 4) [516]. Elevated mitochondrial fission activates autophagy to promote the survival of HCC cells through increased ROS production in in vitro and in vivo models (Table 4) [516]. The autophagic degradation of HIF-1α was reported to be repressed by α-2 adrenergic receptors (ADRB2) signaling and to promote the survival, disease progression, and sorafenib resistance in patients with HCC (Table 4) [517]. The nutrition deprivation-mediated activation of ketone catabolism and ketolysis in HCC cells was shown to suppress AMPK to inhibit the excess activation of autophagy, thereby promoting tumor growth (Table 4) [518]. Conversely, the p300/CBP-associated factor (PCAF) was reported to promote autophagy to inhibit the growth of HCC cells (Table 4) [519]. Moreover, mitophagy was recently shown to remove p53 to maintain hepatic cancer stem cells (CSCs), thus eliciting hepatocarcinogenesis (Table 4) [520]. These studies strengthen the scenario that autophagy is involved in multiple biological pathways and contributes to the tumorigenesis of HCC.

In addition to HCC, autophagy was indicated to function in the development of cholangiocarcinoma. Autophagy was first shown to be activated to participate in vitamin K2 (a well-known antitumor agent for treating malignant leukemia and HCC)-induced cell growth inhibition in cholangiocellular carcinoma cell lines (Table 4) [521]. In addition to vitamin K2, autophagy was reported to be induced in cholangiocellular carcinoma cell lines and mouse xenografts treated with decitabine, an inhibitor of DNA methyltransferase that was shown to suppress tumor development of various cancer types (Table 4) [522]. Activated autophagy was detected in tumor cells from a xenograft tumor model in nude mice and in specimens from clinical cholangiocellular patients (Table 4) [523]. Interference with autophagy was demonstrated to induce cell apoptosis in cholangiocellular cell lines, to suppress cholangiocellular tumor development in a mice xenograft model, and to enhance chemotherapy sensitivity of cisplatin-treated cholangiocellular cells [523]. Moreover, a lower Beclin 1 expression was shown to be associated with lymph node metastasis and poor survival rates of intrahepatic cholangiocellular carcinoma (IHCC) patients (Table 4) [524,525]. Analogously, autophagy was also activated in Kras^G12D^ mutations and p53 deletions (Kras/p53)-induced IHCC in mice (Table 4) [526,527]. The blockade of autophagy by CQ inhibited the tumor cell growth in Kras/p53 IHCC (Table 4) [526,527]. Similarly, CQ was shown to increase the chemosensitivity of cisplatin-treated human cholangiocarcinoma (CCA) cells (Table 4) [528] and to enhance cell apoptosis of human CCA cells through ER stress (Table 4) [529]. In addition, the expressions of HIF-1α, BNIP3, and PI(3)KC3 were positively correlated with lymph node metastasis and poor survival rate in cholangiocellular carcinoma patients [530]. The expressions of LC3B, Beclin 1, and p62/SQSTM1 was shown to be increased at the early stage of multistep cholangiocarcinogenesis in hepatolithiasis [531]. Autophagy was also induced by 5-fluorouracil (5-FU)-treated CCA cells (Table 4) [532]. The 5-FU-activated autophagy was inhibited by capsaicin (the major pungent ingredient found in hot red chili peppers that was shown to repress cell growth of malignant tumors) (Table 4) [532], suggesting that autophagy may participate in the multidrug resistance of chemotherapy in CCA. Treatments with the sphingosine kinase 2 inhibitor, ABC294640, in CCA cells was shown to induce autophagy, and the inhibition of autophagy by CQ and BAF-A1 potentiated ABC294640-induced cell cytotoxicity and apoptosis (Table 4) [533]. The suppressed autophagic flux was indicated to contribute to oblongifolin C (a natural, small molecule that induces cell apoptosis in cervical cancer)-induced cell apoptosis of human CCA cells (Table 4) [534] and to salinomycin-inhibited tumor cell growth in CCA mouse xenograft model (Table 4). Additionally, p53 was demonstrated to activate autophagy in CCA cells, and the repression of p53 was reported to enhance the chemosensitivity in nutrient-deprived CCA cells through downregulating autophagy (Table 4) [535]. In line with this study, compound C (a pharmacological inhibitor of AMPK) treatment in human CCA cells was shown to induce p53-dependent autophagy to protect cells from apoptosis (Table 4) [536]. On the contrary, autophagy was indicated to participate in microRNA (miR)-124-induced cell death in human CCA cells (Table 4) [537] and in dihydroartemisinin-triggered cell death of human CCA cells through the death-associated protein kinase (DAPK)–Beclin 1 pathway (Table 4) [538]. Collectively, these studies indicate that autophagy may participate in the development of IHCC, and the modulation of autophagy by pharmacological inhibitors could serve as a therapeutic option to treat IHCC.

### 4.4. Viral Hepatitis

In the past decade, hepatic autophagy has been extensively shown to participate in the liver cells infected with hepatitis viruses, including HBV and HCV. In the late 2000s, the HCV-H77 (genotype 1a) infection was first shown to activate the formation of autophagic vacuoles in immortalized human hepatocytes (IHH) (Table 5) [541]. Furthermore, in the replication of a full-length HCV-JFH1 (genotype 2a) viral RNA in human HCC, Huh7 cells were found to induce an incomplete autophagic process that enhanced the accumulation of autolysosomes and interfered with autophagic flux and degradation (Table 5) [542]. HCV-JFH1-induced autophagosome formation was further demonstrated to be required for viral RNA replication (Table 5) [542]. Moreover, ATG5 was shown to interact with the nonstructural (NS) proteins of HCV, NS4B (a viral protein that reconstitutes a membranous web for the replication of viral RNA), and NS5B (a RNA polymerase that replicates viral RNA) to promote HCV viral replication (Table 5) [543]. The HCV-induced autophagosomal membrane was reported to provide a membraned platform that contains HCV NS5A, NS5B, and viral RNA for replication (Table 5) [544,545,546]. These studies imply that HCV activates the autophagic process to organize a membranous structure for replicating viral RNA. However, several contradictory studies have indicated that HCV-induced autophagic vacuoles do not colocalize with viral proteins or viral RNA (Table 5) [547,548,549,550,551], suggesting that the viral-triggered biogenesis of autophagic vacuoles may not be necessary for viral RNA replication.

The infection by a cell culture-derived HCV-JFH1 virus (HCVcc) into Huh7.5.1 cells, a clone derived from Huh7 cells that is highly permissive to HCV growth, is known to promote the translation of incoming viral RNA, thus establishing a virus infection (Table 5) [548]. Another study showed that a HCVcc-JFH1 infection activates an entire autophagic process throughout the maturation of autolysosomes to support viral RNA replication by suppressing the HCV pathogen associated molecular pattern-mediated innate antiviral responses (Table 5) [548,549]. Similarly, the genetical silencing of ATGs, including Beclin and ATG7, was found to suppress HCV viral infectivity and to upregulate interferon-stimulated gene (ISG) expression in HCV H77-infected IHH cells (Table 5) [552]. Moreover, a recent study revealed that HCV may activate autophagy to degrade the tumor necrosis factor receptor (TNFR)-associated factor 6 (TRAF6) through p62/SQSTM1, thus alleviating innate host immunity (Table 5) [553]. These results together indicate that HCV may induce hepatic autophagy to repress innate antiviral immunity to promote the HCV-infecting life cycle in liver cells (Table 5) [548,549,552,554]. Moreover, the HCV infection was shown to participate in the assembly of infectious particles rather than to affect the replication of intracellular viral RNA (Table 5) [555]. Further studies have demonstrated that autophagy mediates the egress of HCV virion through the CD63-associated exosome pathway (Table 5) [556] and the regulation of apolipoprotein E (ApoE) transport (Table 5) [557]. Additionally, HCV-activated autophagy may depend on different HCV genotypes (Table 5) [558]. The replication of HCV-Con1 (genotype 1b) was reported to impede the fusion of autophagosomes with lysosomes, thus protecting infected liver cells from cell death (Table 5) [558].

Numerous studies have demonstrated that the ER stress-associated unfolded protein response (UPR) is required for HCV-induced autophagy (Table 5) [542,548,549,554], presumably through the DNA damage-inducible transcript 3 protein (DDIT3, also known as CHOP)-mediated transcription of LC3B and ATG5, to activate autophagy (Table 5) [559]. Furthermore, UPR is required for interference with protein kinase B (PKB)-tuberous sclerosis (TSC)-mTOR complex 1 (mTORC1) signaling (Table 5) [560]. STX17, an autophagy regulator implicated in the biogenesis of autophagic vacuoles, may regulate HCV virion egress (Table 5) [561] and promote viral RNA replication (Table 5) [562]. Recently, the alternative splicing of ATG10 was shown to regulate HCV replication through affecting autophagic flux (Table 5) [563,564].

Apart from bulk and nonselective autophagy, HCV was shown to activate autophagy to selectively degrade intracellular organelles, such as LDs and mitochondria (Table 5) [430,565]. The level of lipidated LC3 was inversely correlated with the clinical parameters related to hepatic steatosis in the liver samples of patients with chronic HCV (Table 5) [430]. The HCV-activated autophagic vacuoles were found to engulf LDs in HCV-replicon and HCV-infected cells (Table 5) [430]. In addition, interference with autophagy by pharmacological inhibitors and gene knockdown was demonstrated to significantly enhance the intracellular levels of cholesterol (Table 5) [430], suggesting that the inhibition of autophagic degradation of LDs may result in the occurrence of hepatic steatosis in chronically HCV-infected patients. These results indicate that HCV could induce autophagy to catabolize LDs to protect HCV-infected cells against excess LDs accumulation. Moreover, HCV infection has been shown to trigger the formation of mitophagosomes that degrade mitochondria via the PINK1–Parkin axis in infected cells, thereby promoting HCV viral RNA replication (Table 5) [565,566]. Another study indicated that HCV-induced mitochondrial fission is required for the elimination of mitochondria in infected cells (Table 5) [567]. These studies argue that HCV may activate autophagic degradation to clear the mitochondria with viral-induced damage to attenuate the cell apoptosis of infected liver cells (Table 5) [567]. Oppositely, another study reported that HCV capsids interact with Parkin to impede the translocation of Parkin from the cytoplasm to mitochondria to inhibit mitophagy and to sustain HCV-induced mitochondrial damage and injury in infected liver cells (Table 5) [568]. Recently, HCV viral RNA was shown to induce selective autophagy, possibly promoting the elimination of ubiquitinated protein aggregates (Table 5) [569]. However, CMA was also reported to be activated by HCV and implicated in the degradation of IFN-alpha receptor-1 (IFNAR1), which is stimulated by FFAs (Table 5) [570]. Another study demonstrated that HCV NS5A interacts with HSC70 to promote the degradation of hepatocyte nuclear factor 1α (HNF1α) (Table 5) [571]. Recently, IFN-β-inducible SCOTIN was shown to recruit HCV NS5A into autolysosomes for degradation, restricting the HCV infection (Table 5) [572]. These studies collectively suggest that HCV subverts autophagy to antagonize viral RNA replication and the harmful host responses in infected liver cells.

The induction of autophagy by HBV was first discovered through observing the autophagic degradation of peroxisomes in the liver tissues of patients with chronic HBV who received kidney transplantations and immunosuppressive therapy [329]. Levine et al. showed that the heterozygous deletion of Beclin 1 downregulates autophagy in liver cells, thus promoting HBV-associated premalignant lesions [573]. The HBV infection was further revealed to induce the early-stage formation of autophagic vacuoles in Huh7.5.1 cells through increasing PI(3)K enzyme activity to promote HBV DNA replication (Table 5) [574,575]. The role of autophagy in HBV DNA replication was further confirmed by a study that revealed that the ablation of ATG5 inhibits autophagy to reduce the HBV gene expression in liver cells and in sera (Table 5) [576]. HBV viral proteins, such as HBV x protein (HBx) and HBV small surface protein (SHB), activate autophagy in liver cells through enhancing the Beclin 1 expression and inducing UPR (Table 5) [577,578], thus promoting HBV replication. Recently, HBV was shown to subvert the ATG5-ATG12-ATG16L complex rather than the lipidated ATG8/LC3-mediated autophagosome biogenesis to support the assembly and stability of the HBV nucleocapsid, suggesting that HBV exploits nondegradative autophagy to facilitate HBV propagation (Table 5) [579]. Moreover, HBV can promote autophagy through the miR-192-3p-XIAP axis to benefit HBV replication in vitro and in vivo (Table 5) [580]. In addition to these findings that autophagy promotes HBV growth, Epigallocatechin-3-gallate (EGCG), a major polyphenol in green tea, was demonstrated to inhibit HBV-induced autophagy through increasing lysosomal acidification, thus repressing HBV growth (Table 5) [581]. In spite of genomic DNA replication, HBV-activated autophagy may participate in the secretion of infectious virions in infected cells (Table 5) [582].

HBx-induced autophagy was reported to inhibit mitochondrial apoptosis to promote the survival of HBV genomic DNA-transfected cells (Table 5) [583]. Also, autophagy and ER-associated degradation (ERAD) were shown to promote the degradation of HBV enveloped proteins for the establishment of chronic HBV infection (Table 5) [584]. Additionally, autophagy was negatively correlated with an oncogenic miR-244, which was degraded through the autophagic process in the liver tissues of HBV-associated patients with HCC and HBx-transgenic mice (Table 5) [585]. Reduced ATG5 expression and elevated miR-244 level were significantly correlated with HBV infection and a poor survival rate (Table 5) [585], suggesting that autophagy promotes the degradation of miR-244 to repress HBV-associated tumorigenesis. HBx was shown to activate autophagy through the PI(3)K/Akt-mTOR pathway [586], death-associated protein kinase [587], the JNK2-mediated regulation of Beclin 1/Bcl-2 interaction [588], and the upregulation of HMGB1 [589] (Table 5). By contrast, the transfection of HBV genomic DNA and overexpression of HBx in hepatoma cells interfere with autophagic degradation by impairing the maturation of lysosomes (Table 5) [590]. Additionally, accumulated p62/SQSTM1 and defected cathepsin D were positively related to the liver tissues of people with chronic HBV infection and HBV-associated liver cancer (Table 5) [590], suggesting that HBV’s inhibition of autophagy could result in the development of HCC. Taken together, these results indicate that HBV may modulate autophagy to promote viral growth and may participate in the development of HBV-related liver cancer.

### 4.5. Fibrosis and Cirrhosis

Autophagy was previously shown to eliminate the mutant α1-ATZ aggregated proteins in liver cells [353,354,370]. The enhancement of autophagy by carbamazepine (CBZ) was reported to reduce the accumulation of α1-ATZ and the inhibited hepatic fibrosis in a α1-AT-deficient mouse model (Table 6) [594] and to reduce the death of hepatic cells in fibrinogen-storage-related diseases (Table 6) [595]. Furthermore, activation of autophagy by gene transfer of the autophagy regulator, TFEB, was demonstrated to enhance the removal of α1-ATZ inclusion bodies and to reverse α1-ATZ-induced liver damage in a α1-AT-deficient mouse model (Table 6) [391]. The induction of MAP1s-mediated autophagy by spermidine was found to prevent liver fibrosis and HCC development in mice (Table 6) [596]. Moreover, in a tetrachloride (CCl_4_)-induced fibrosis mouse model, human palatine tonsil-derived mesenchymal stem cells (T-MSCs) that differentiated into hepatocyte-like cells were found to ameliorate hepatic fibrosis in an autophagy-dependent manner (Table 6) [597]. Interference with autophagic degradation by chloroquine was reported to improve CCl_4_-induced liver fibrosis through inhibiting the activation of hepatic stellate cells (Table 6) [598]. The impairment of autophagy by dihydroceramide promoted the disease progression of hepatic steatosis to liver fibrosis (Table 5) [599]. Similarly, impaired autophagy-mediated inhibition of YAP degradation also resulted in liver fibrosis (Table 6) [515]. Autophagy was shown to protect liver sinusoidal endothelial cells (LSECs) from oxidative stress to ameliorate liver fibrosis (Table 6) [600]. These studies collectively indicate that autophagy prevents the development of liver fibrosis.

HIF-1α, ROS-JNK1/2-dependent activation, and the X-box binding protein 1 (XBP1) signaling of UPR were required for the autophagy-mediated activation of hepatic stellate cells (Table 6) [601,602,603]. Moreover, autophagy deficiency was reported to induce cell death in hepatocytes, leading to liver inflammation, hepatic fibrosis, and tumorigenesis by substantially activating Nrf2 (Table 6) [604]. Taken together, these studies indicate that autophagic degradation plays a critical role in preventing liver fibrosis. In addition, macrophages derived from mice with an ATG5 deletion was found to secrete high levels of ROS-induced interleukin 1A (IL-1A) and IL-1B, which were correlated with the enhanced accumulation of matrix and fibrogenic cells in CCl4-induced fibrosis livers and upregulated profibrogenic genes expressions in the cocultured hepatic myofibroblasts (Table 6) [605], suggesting that the activation of autophagy in macrophages may repress liver fibrosis. These studies imply that autophagy regulates multiple biological processes to mitigate the occurrence of hepatic fibrosis.

By contrast, one study has shown that autophagy promotes hepatic fibrogenesis by activating hepatic stellate cells in the CCl4 or thioacetamide (TAA)-induced liver fibrosis mouse model (Table 6) [606]. Similarly, dimethyl α-ketoglutarate (DMKG) was reported to inhibit CCl4-induced hepatic fibrosis through reducing autophagy (Table 6) [607]. Furthermore, quercetin attenuated bile duct ligation (BDL) and CCl4-induced liver fibrosis in mice through inhibiting hepatic stellate cell activation and autophagy (Table 6) [608]. Analogously, 3-MA was reported to ameliorate hepatic fibrosis through impairing autophagy initiation in hepatic stellate cells (Table 6) [609]. The genetic deletion of Golgin A2 (GOLAG2), a cis-Golgi protein, reportedly activates autophagy and promotes liver fibrosis in mice (Table 6) [610]. Furthermore, through activating autophagy, β-arrestin 1 (β-arr1) has been implicated in improving the compensatory proliferation of hepatocytes and the growth of hepatic stellate cells with a consequence in liver fibrosis (Table 6) [611]. These studies indicate autophagy’s promoting role in liver fibrosis development.

Unsurprisingly, autophagy was also correlated with cirrhosis in the liver samples of patients with chronic liver disease [612]. The aggregation of p62/SQSTM1 through defective autophagy was found to be increased in the damaged bile ducts of primary biliary cirrhosis (PBC) patients (Table 6) [613]. The elevated expressions of mitochondrial antigens, including pyruvate dehydrogenase complex-E2 component (PDC-E2) and cytochrome c oxidase, subunit I (CCO), was further shown to be associated with the autoimmune pathogenesis of bile duct lesions in patients with PBC (Table 6) [614]. The increased level of LC3B, the colocalization of LC3B with LAMP1, and the elevated expressions of LAMP2 and cathepsin D were detected in the liver samples of patients with cirrhosis (Table 6) [615]. The impairment of autophagy by CQ was reported to be attenuated the CCl4-induced ductular reaction and fibrosis (Table 6) [615]. By contrast, autophagy was reported to contribute to the formation of hepatocyte growth factor (HGF)-directed Axin2+CD90+ CSCs to promote hepatocarcinogenesis in cirrhotic livers (Table 6) [616]. These studies together constitute the notion that autophagy may regulate the occurrence of liver cirrhosis.

## 5. Conclusions and Future Directions

Autophagy plays a critical role in the maintenance of cellular homeostasis and the elimination of unwanted intracellular materials, such as proteins and organelles, in liver physiology. Autophagy has been extensively shown to be regulated in liver cells in multiple liver diseases. The activation of autophagy has mostly been conceptualized as a protector of liver cells against injury. Furthermore, it has been suggested to prevent the development of liver-associated diseases through the autophagic degradation of aggregate-prone proteins and damaged organelles. In accordance with the protective roles of autophagy, the enhancement of autophagy in liver cells is considered a treatment strategy for ameliorating the development of liver diseases. By contrast, autophagy has also been implicated in promoting liver damage-induced cell death and the development of liver-related diseases, suggesting that interference with autophagy may represent a novel route to mitigate the progression of liver diseases. Hence, the physiological significance of autophagy in liver diseases is still debated and is largely hampered by discrepancies between studies. Our current understanding of hepatic autophagy in liver diseases seems to be context-dependent and disease stage-dependent. Nevertheless, further investigations are required to comprehensively dissect the detailed role of autophagy in the induction and progression of liver-related diseases. Moreover, an improved understanding of the clinical relevance of autophagy in the different stages of liver disease progression is required. Furthermore, the innovative and integrative system-based biology approach should be employed to identify distinct aspects of autophagy that can be targeted and used for the development of a feasible treatment strategy to effectively cure liver diseases in clinical settings.

## Figures and Tables

**Figure 1 ijms-20-00300-f001:**
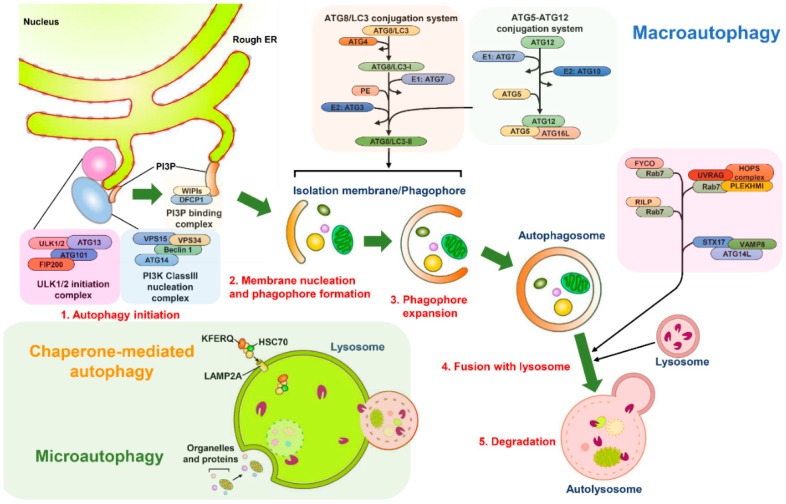
Schematic diagram of the autophagy pathway. There are three main types of autophagy: microautophagy, chaperone-mediated autophagy (CMA), and macroautophagy. The process of microautophagy undergoes an invagination and scission process of the lysosomal membrane that sequestrate the cytosolic portions into the lysosomal lumen for degradation. In CMA, the heat-shock cognate protein of 70 kDa (HSC70) recognizes the substrates that contain the pentapeptide “Lys-Phe-Glu-Arg-Gln” (KFERQ) motifs and deliver them to lysosomes through interacting with lysosomal membrane protein 2A (LAMP2A). Macroautophagy is a stepwise vacuole biogenesis process that initiates with the nucleation of the membrane to form a phagophore, the expansion of a phagophore to the closure of autophagosomes, and the fusion of autophagosomes with lysosomes to mature autolysosomes. Finally, the engulfed intracellular components are eliminated within the autolysosomes. When the cells are deprived of nutrients, the inhibition of the mammalian target of rapamycin (mTOR) complex induces the translocation of the unc-51 like-kinase (ULK1/2) complex (ULK1/2, autophagy-related gene (ATG) 13, RB1-inducible coiled-coil 1 (RB1CC1, also known as FIP200), and ATG101) to the membrane nucleation site. The translocated ULK1/2 complex in turn recruits and activates the class III phosphatidylinositol-3-OH kinase (class III-PI(3)K complex, including Vps34/PI(3)KC3, Vps15, Beclin 1, and ATG14) to produce phosphatidylinositol-3-phosphate (PtdIns(3)P). Subsequently, PtdIns(3)P recruits the double-FYVE-containing protein 1 (DFCP1) and WD-repeat domain PtdIns(3)P-interacting (WIPI) family proteins to promote the formation of the isolation membrane (IM)/phagophore. Two ubiquitin-like (UBL) conjugation systems underlie the expansion and elongation of the phagophore to form mature autophagosomes. The ubiquitin conjugation enzyme 1 (E1) ATG7 activates ATG12 through a thioester bonding with ATG12, and then ATG12 is transferred to ATG10 enzyme 2 (E2). ATG12 is finally conjugated to ATG5, forming an ATG5-ATG12 complex, which in turn interacts with ATG16L to form an ATG12-ATG5-ATG16L complex. To successfully conjugate phosphatidylethanolamine (PE) to ATG8/LC3 family proteins, the ATG8/LC3 family proteins are cleaved by the cysteine protease ATG4 to expose the C-terminal glycine residues, generating the ATG8/LC3-I. Then, ATG8/LC3-I is covalently linked to PE to form the lipidated form of LC3 (ATG8/LC3-II) via an enzyme cascade of ATG7 E1 and ATG3 E2. The fusion of autophagosomes and lysosomes relies on the interactions between the small GTPase Ras-related protein 7 (Rab7) and cytoskeleton-associated factors, the FYVE and coiled-coil domain-containing 1 (FYCO1) and Rab-interacting lysosomal protein (RILP). Additionally, UV radiation resistance-associated (UVRAG), the homotypic fusion and protein sorting (HOPS) complex, pleckstrin homology domain-containing protein family member 1 (PLEKHM1), and protein complex containing syntaxin 17 (STX17), vesicle-associated protein 8 (VAMP8), and synaptosome-associated protein 29 (SNAP29) are involved in the maturation process of autolysosomes.

**Figure 2 ijms-20-00300-f002:**
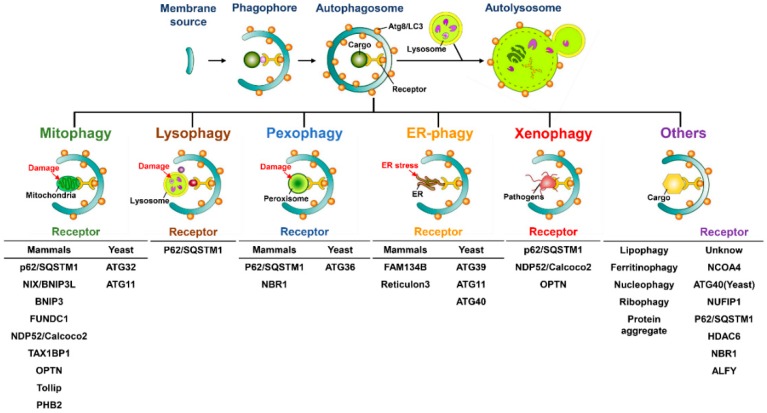
Different modes of selective autophagy. Degradation of selective autophagy involves the specific interactions between the ATG8/LC3-interacting regions (LIRs) within cargo receptors and ATG8/LC3 located onto autophagosomal membrane. The ubiquitination of degradative cargos or associated ligand proteins is often required for recognition by cargo receptors. Selective autophagy has been shown to eliminate different kinds of organelles and proteins, including damaged mitochondria (mitophagy), injured lysosomes (lysophagy), damaged peroxisomes (pexophagy), stressed ER (ER-phagy), and infectious pathogens (xenophagy), through specific cargo receptors in yeast and mammalian cells as indicated. Also, lipid droplets (LDs) ferritin, nuclei, ribosomes, and protein aggregates are also degraded by selective autophagy via the identified cargo receptors and other unknown adaptor proteins.

**Table 1 ijms-20-00300-t001:** Summary of autophagy in liver physiology.

Experimental Model	Characteristics of Autophagy	Function of Autophagy	References
Rat liver (Perfusion of glucagon)	Electron micrograph of polymorphic dense bodies	Sequestration of deformed mitochondria that is associated with glucagon-related catabolic process	[14]
Rat liver (Intravenous injection of Triton WR-1339)	Electron micrograph of polymorphic dense bodies that includes single- and double-membraned vesicles (termed cytolysomes)	Degradation and degeneration of mitochondria in Triton-treated hepatic cells	[15]
Rat liver (Intravenous injection of glucagon)	Biochemical fractionation of lysosomes and autophagic vacuoles	Association with lysosomes and enrichment of lysosomal acidic enzymes within autophagic vacuoles for protein degradation in the glucagon-stimulated liver cells	[19]
Rat liver (Intravenous injection of glucagon)	Biochemical fractionation of lysosomes and autophagic vacuoles Electron micrograph of autophagic vacuoles that engulf organelles	Increased number of autophagic vacuoles by glucagon stimulation Increased lysosomal proteases in glucagon-treated hepatic cells Degradation of intracellular organelles in hepatic cells treated by glucagon	[20]
Rat liver (Intravenous injection of glucagon)	Biochemical fractionation of lysosomes and autophagic vacuoles Electron micrograph of single- and double-membranous autophagic vacuoles that sequestrate organelles	Increased formation of autophagic vacuoles by glucagon-treated liver cells Increased lysosomal proteases in autophagic vacuoles in hepatic cells treated by glucagon Degradation of intracellular organelles in glucagon-treated liver cells	[270]
Rat liver (Intraperitoneal injection of dimethylnitrosamine (DMNA))	Electron micrograph of autophagic vacuoles Increased autophagic vacuoles when the onset of necrosis is detected	Degradation of organelles by autophagic vacuoles in DMNA-treated liver cells Associated with the occurrence of hepatocellular necrosis	[272]
Rat liver (Long-term starvation)	Electron micrograph of autophagic vacuoles that engulf organelles	Degradation of organelles by autophagic vacuoles in hepatocytes of liver atrophy induced by starvation Decomposition of glycogen in the starved liver cells of liver atrophy	[273,274,275]
Rat liver (Intraperitoneal injection of glucagon and cycloheximide)	Electron micrograph of autophagic vacuoles	The correlation of autophagic vacuoles formation with the rate of protein synthesis and the level of energy	[276,277,278]
Mouse liver (Intravenous injection of lysine acetylsalicylate	Electron micrograph of single- and multiple-membranous autophagic vacuoles	The engulfment of intracellular components within autophagic vacuoles that may protect the lysine acetylsalicylate-treated liver cells against injury	[279]
Rat liver (Hypothermia)	Electron micrograph of autophagic vacuoles that sequestrate enlarged mitochondria and disorganized endoplasmic reticulum (ER)	The elimination of intracellular organelles by hypothermia-induced autophagic vacuoles in hepatocytes	[280]
Rat liver (Intraperitoneal injection of vinblastine)	Electron micrograph of autophagic vacuoles that engulf organelles	Degradation of intracellular organelles within autophagic vacuoles	[281]
Rat liver (Perfusion of amino acids-deprived medium)	Electron micrograph of autophagic vacuoles that engulf organelles	Maintenance of intracellular amino acids in hepatocytes and recycling of nutrients	[21]
Rat liver (Perfusion of amino acids-deprived medium)	Electron micrograph of autophagic vacuoles that engulf organelles	Decreased the intracellular amounts of glucogenic amino acids by autophagy	[25]
Rat isolated hepatocytes Mouse liver (Starvation and refeeding)	Electron micrograph of autophagic vacuoles that engulf organelles Protein degradation of endogenous proteins	Degradation of intracellular organelles and endogenous proteins, which is inhibited by methylated adenosine derivatives and refeeding	[24,30,282]
Rat liver	Electron micrograph of autophagic vacuoles that engulf organelles Isolation of autophagic vacuoles Inhibited formation of autophagic vacuoles by insulin	Degradation of intracellular organelles and endogenous proteins, which is inhibited by insulin, 3-methyladenine (3-MA), vinblastine, and amino acids	[22,23,287,288,289,290,291,292,293,294]
Newborn rat hepatocytes	Electron micrograph of autophagic vacuoles that is closely related to the degradation of fetal-type glycogen	Degradation of fetal-type glycogen in the neonatal period	[295]
Rat liver (Perfusion of orotic acid) Isolated rat hepatocytes		Autophagic degradation of RNA and proteins in liver, which is inhibited by chloroquine and amino acids	[296]
Rat liver	Electron micrograph and biochemical fractionation of lysosomes and autophagic vacuoles	Degradation of proteasomes by lysosomes and autophagic vacuoles	[297]
Rat liver		Restriction of ischemic liver injury by inhibition of autophagy	[298]
Rat liver Isolated rat hepatocytes		Enhancement of cell survival of carcinogen-treated hepatocytes by reduced autophagy	[299]
Primary rat hepatocytes	Electron micrograph of autophagic vacuoles	Increased the intracellular iron pool by autophagic turnover of ferritin and iron-containing proteins	[300]
Liver specimens of patients	Electron micrograph of autophagic vacuoles	Promotion of cell death in anorexia nervosa livers of patients by starvation-induced autophagy	[301]
Rat hepatoma H4IIE cells		Support of amino acids from autophagic proteolysis of endogenous proteins to the regulation of translational effectors	[302]
Wild type and liver-specific knockout of ATG7 mice	Electron micrograph of autophagic vacuoles Detection of lipidation of ATG8/LC3	Maintenance of blood glucose and amino acids levels by hepatic autophagy	[303]
Primary human and mouse hepatic stellate cells	Analysis of autophagic flux by the fluorescent signal of mRFP-GFP-LC3 Detection of changes in autophagic flux by interference with autolysosome maturation	Involvement of enhanced autophagic flux in the activation of hepatic stellate cells	[304]
Rat liver	Degradation of cathepsin family enzymes (B, H, and L)	Reduced lysosomal proteolysis by suppression of autophagy in regenerating liver	[305]
Rat liver	Immunofluorescence analysis of ATG8/LC3 Electron micrograph of autophagic vacuoles	Involvement of autophagy in the degeneration of hepatocytes of liver grafts	[306]
Wild type and liver-specific Tet-off-LAMP2A transgenic mice	Electron micrograph of CMA-mediated autophagic process	The maintenance of liver function and protection against liver damage by hepatic CMA	[79]
Mouse hepatocytes (In vivo and in vitro ischemia and reperfusion)	Detection of the lipidation of ATG8/LC3 Analysis of autophagic flux by fluorescence signal of AdmCherry-GFP-LC3 Detection of the processing and expression of cathepsin D	Amelioration of liver damage and restoration of mitochondrial function in liver after ischemia and reperfusion	[307]
Human hepatoma, Huh7 cells Wild type and keratin 8-overexpressing mice	Detection of the lipidation of ATG8/LC3 Colocalization of p62/SQSTM1 with keratin 8 and GFP-LC3 with HSP70 Electron micrograph of autophagic vacuoles	Elimination of components of MDBs by rapamycin-induced autophagy	[308,309]
Wild type and liver-specific knockout of ATG7 mice Rat hepatocyte RALA255-10G Primary rat hepatocytes Human hepatoma, Huh7 and HepG2 cells	Electron micrograph of autophagic vacuoles that engulf LDs Detection of the lipidation of ATG8/LC3 Colocalization of ATG8/LC3 with LDs-associated proteins Biochemical fractionation of autophagic vacuoles that contain LDs-related proteins	Involvement of ATG8/LC3 conjugation system in the biogenesis of LDs Promotion of LDs catabolism by hepatic autophagy, i.e., “lipophagy” Degradation of lipid by hepatic autophagy Degradation of Apolipoprotein B located around LDs	[243,244,310,311]
Wild type and liver specific knockout of ATG7, VHL, HIF-1α, and HIF-2α mice Rat liver (Antilipolytic agents treatment) Isolated rat hepatocytes	Detection of GFP-LC3-labeled autophagic vacuoles that sequestrate peroxisomes Electron micrograph of autophagic vacuoles that engulf peroxisomes	Selective engulfment of excess peroxisomes by autophagy (termed “pexophagy”) Degradation of peroxisomes by HIF-2α-mediated selective autophagy Decreased peroxisomal enzyme activities by enhanced degradation of peroxisomes by autophagy, which is inhibited by 3-MA	[312,313,314,315,316,317,318]
Primary rat hepatocytes Isolated hepatocytes from wild type and GFP-LC3-transgenic mice Wild type and Vps34/PI-3KC3^D761A^ mice Rat liver	Electron micrograph of autophagic vacuoles that engulf mitochondria Translocation of mitochondria into acidic organelles Engulfment of mitochondria by GFP-LC3-labeled autophagic vacuoles in the nutrient starved- and glucagon-treated hepatocytes Mildly inhibited hepatic autophagy by inactivation of Vps34/PI-3KC3	The remodeling of mitochondria by mitochondrial autophagy (termed “mitophagy”) The degradation of mitochondria by hepatic mitophagy Restricted mitochondrial respiration and gluconeogenesis by inactive Vps34/PI-3KC3-interfered mitophagy Accumulated injured mitochondria by age-related failure of mitophagy	[319,320,321,322,323]
Mouse hepatoma, Hepa1-6 cells Wild type, ATG5^−/−^, and ATG7^−/−^ mouse embryonic cells (MEF) Ob/ob mice and high fat diet (HFD)-induced mice Wild type and liver-specific knockout of ATG7 mice	Downregulation of hepatic autophagy by obesity in ob/ob mice and HFD-mice Electron micrograph of autophagic vacuoles that contains ER	Induced insulin resistance and ER stress by obesity-inhibited hepatic autophagy Selective modulation of UPR by hepatic autophagy	[321,324]

**Table 2 ijms-20-00300-t002:** Summary of autophagy in liver injury.

Experimental Model	Characteristics of Autophagy	Function of Autophagy	References
Rat liver (Intraperitoneal injection of DMNA)	Electron micrograph of autophagic vacuoles Increased autophagic vacuoles when the onset of necrosis is detected	Degradation of organelles by autophagic vacuoles in DMNA-treated liver cells Associated with the occurrence of hepatocellular necrosis	[272]
Mouse liver (Intravenous injection of lysine acetylsalicylate	Electron micrograph of single- and multiple-membranous autophagic vacuoles	The engulfment of intracellular components within autophagic vacuoles that may protect the lysine acetylsalicylate-treated liver cells against injury	[279]
Rat liver (Stressors: fasting, cortisol injection, reserpine injection, restraint, spinal cord transection, etc.)	Electron micrograph of single- and multiple-membranous autophagic vacuoles	Protection of liver cells against multiple stress responses	[364]
Rat liver (Lethal *Escherichia coli*)	Electron micrograph of autophagic vacuoles	Association of hepatic autophagy in *Escherichia coli*-induced liver injury	[365]
Rat hepatocytes (Calcium ionophore, microtubule active agents, and hepatotoxins)	Detection of autophagic degradation of endogenous proteins	Decreased autophagic degradation by liver injury	[366]
Liver specimens of normal and α1-AT-deficeint patients Wild type and PiZ (α1-AT-deficeint) mice	Electron micrograph of autophagic vacuoles that engulf mitochondria Colocalization of α1-AT and cathepsin D in fasted mouse liver Enhanced the formation of autophagic vacuoles in hepatocytes of fasting mouse liver Sequestration of ER-retained and mutated α1-ATZ aggregated protein by autophagic vacuoles	Induced mitochondrial injury and mitochondrial autophagy in α1-AT-deficient livers Suppressed autophagy activation of deficiency of α1-AT in liver Autophagic elimination of α1-ATZ aggregated protein Reduced the inclusion bodies of α1-ATZ aggregated protein and liver injury by rapamycin-induced hepatic autophagy	[353,354,355,371,372,373]
Wild type and liver-specific knockout of ATG7 mice Hepatocellular carcinoma cell lines	Impaired autophagy by ATG7-deficicent mice by electron microscopy analysis and detection of LC3 lipidation Degradation of p62/SQSTM1 by autophagy	Induced hepatomegaly and the swelling of hepatic cells deficient of ATG7 Accumulated peroxisomes and deformed mitochondria in the liver cells of ATG7-deficient mice Accumulated ubiquitin- and p62/SQSTM1 containing inclusion bodies in the liver cells of ATG7-deficient mice Suppression of liver dysfunction in ATG7-deficient mice by additional knockout of p62/SQSTM1 Formation of p62/SQSTM1- and Keap1-postivie inclusion bodies in ATG7-deficient hepatocytes Induction of Nrf2-dependent transcriptions of antioxidant genes in ATG7-deficient livers by p62/SQSTM1-Keap1 interaction Amelioration of liver dysfunction in ATG7-deficient livers by additional ablation of Nrf2 Exacerbation of liver injury in ATG7-deficient livers by additional knockout of Nrf2 Induction of hepatocellular carcinoma in the mice livers loss of ATG7 through activation of Nrf2	[356,377,378,379,380,381]
Liver specimens of liver-related diseases patients	Electron micrograph of autophagic vacuoles in liver after ischemia and reperfusion Enhanced lipidation of ATG8/LC3 Increased of the formation of autophagic vacuoles in the ischemia and reperfusion-injured liver with ischemic preconditioning	Association of increased autophagy with the chemotherapy-injured liver after ischemic preconditioning	[382,383]
Mouse hepatocytes (In vivo and in vitro ischemia and reperfusion)	Detection of the lipidation of ATG8/LC3 Analysis of autophagic flux by fluorescence signal of AdmCherry-GFP-LC3 Detections of the processing and expression of cathepsin D	Amelioration of liver damage and restoration of mitochondrial function in liver after ischemia and reperfusion	[307]
Wild type and GFP-LC3-transgenic mice Isolated primary mouse hepatocytes	Detection of lipidation of ATG8/LC3 and p62/SQSTM1 degradation Electron micrograph of autophagic vacuoles that engulf mitochondria The expression of GFP-LC3-labeled punctate structure	Induction of autophagy in livers of ethanol-treated mice Enhanced cell apoptosis of hepatocytes and liver injury by inhibition of autophagy Activation of mitophagy to modulate ethanol-induced liver steatosis	[345,384]
Primary mouse hepatocytes (LPS; Cecal ligation and performation)	Electron micrograph of autophagic vacuoles Detection of lipidation of ATG8/LC3 Immunofluorescence analysis of ATG8/LC3	Induction of hepatic autophagy through HO-1 upregulation in the hepatocytes treated with LPS and cecal ligation of performation Enhanced cell apoptosis in in the hepatocytes treated with LPS and cecal ligation of performation by interfered with autophagy	[385]
Human hepatoma, HepG2 cells Mouse liver	Electron micrograph of autophagic vacuoles Detection of lipidation of ATG8/LC3 Immunofluorescence analysis of GFP-LC3-labeled punctate structure	Protection of fatty acids-induced lipotoxicity and liver injury by autophagy	[386]
Chang liver cells (PEI treatment)	Electron micrograph of autophagic vacuoles Immunofluorescence analysis of ATG8/LC3	Enhancement of the PEI-induced cytotoxicity in liver cells by autophagy	[387]
Wild type and liver-specific knockout of ATG7 mice (APAP treatment)	Electron micrograph of autophagic vacuoles Detection of lipidation of ATG8/LC3	Induction of hepatic autophagy by APAP treatment Induced cell apoptosis and liver injury in the ATG7-deficient mouse livers treated with APAP Enhanced mitochondrial damage, ROS production, and depolarization of mitochondria in the ATG7-deficient mouse livers treated with APAP	[388]
Rat liver (Cold ischemia/warm reperfusion; liver transplantation)	Electron micrograph of autophagic vacuoles Immunofluorescence analysis of ATG8/LC3	Association of hepatic autophagy with cold ischemia/warm reperfusion-induced cell death of liver injury Suppressed liver injury by inhibition of autophagy Participation of autophagy in the cell death of dysfunctional liver graft	[389]
Liver specimens of patients	Electron micrograph of autophagic vacuoles	Promotion of cell death in anorexia nervosa livers of patients by starvation-induced autophagy	[301]
Wild type and IGF-1 deficient mice Human hepatoma, HepG2 cells	Immunofluorescence analysis of GFP-LC3-labeled punctate structure	Suppressed autophagy in senescent livers Rescued hepatic autophagy by loss of IGF-1 Amelioration of ageing-induced liver injury by IGF-1 deficiency-induced autophagy activation	[390]
Wild type and PiZ (α1-AT-deficeint) mice Wild type and ATG7^−^^/^^−^ MEF	Detections of lipidation of ATG8/LC3 and p62/SQSTM1 degradation Immunofluorescence analysis of GFP-LC3-labeled punctate structure Electron micrograph of autophagic vacuoles that eliminates α1-ATZ inclusion bodies	Enhanced autophagic flux by liver-directed gene transfer of TFEB Increased autophagic degradation of mutated α1-ATZ polymer by liver-directed gene transfer of TFEB Decreased the α1-ATZ-induced liver injury by TFEB-enhanced autophagy	[391]
Isolated hepatocytes from wild type and caspase 1^−^^/^^−^ mice	Immunofluorescence analysis of GFP-LC3-labeled punctate structure Detection of lipidation of ATG8/LC3	Decreased autophagic flux in caspase 1^−^^/^^−^ hepatocytes after hypoxia/reoxygenation Activation of autophagy through caspase 1-dependent upregulation of Beclin 1 Selective elimination of mitochondria by caspase 1-mediated autophagy activation Reduced liver injury and oxidative stress after hypoxia/reoxygenation by caspase 1-mediated autophagy	[392]
Isolated mouse hepatocytes and Kupffer cells	Detection of lipidation of ATG8/LC3	Enhanced the activation of NLRP3 inflammasome after LPS challenge by deficiency of Kir6.2/K-ATP channel Aggravated ER stress and autophagy in the Kir6.2-deficicency-induced NLRP3 inflammasome activation	[393]
Mouse liver	Detections of lipidation of ATG8/LC3 and p62/SQSTM1 degradation	Induction of APAP-induced hepatic necrosis by protein kinase C Suppressed APAP-induced hepatic cytotoxicity by PKC inhibitors Repressed APAP-induced hepatic cytotoxicity by activated autophagy	[394]
Isolated mouse hepatic stellate cells	Detections of lipidation of ATG8/LC3 and p62/SQSTM1 degradation	Decreased hepatic stellate cell activation and autophagy by blocking of inositol-requiring enzyme 1 (IRE1α) Induced the oxidative stress and Nrf2-dependent antioxidant response by IRE1α blocking-repressed autophagy	[395]
Rat liver (Ischemia/reperfusion)	Detections of lipidation of ATG8/LC3	Alleviated ischemia/reperfusion-induced liver injury by tunicamycin-preconditioning ER stress The repressed ischemia/reperfusion-induced liver injury by IRE1 and glucose-regulated protein 78 Prevented live apoptosis after ischemia/reperfusion by activated autophagy	[396]
Mouse liver (LPS/D-galactosamine (GalN))	Detections of lipidation of ATG8/LC3	Induced ER stress and autophagy at the early stage of LPS/GalN-induced liver injury	[397]
Mouse liver (Ischemia/reperfusion; fasting)	Detections of lipidation of ATG8/LC3	Protection of hepatic injury after ischemia/reperfusion by one-day fasting Reduced circulating HMGB1 and cytoplasmic translocation of HMGB1 and activated autophagy in liver injury by fasting Suppressed protection of liver injury by inhibition of autophagy Prevented HMGB1 translocation and elevated circulating HMGB1 by Sirt1 inhibition	[398]
Wild type and liver-specific knockout of ATG7 mice	Detections of lipidation of ATG8/LC3 and p62/SQSTM1 degradation	Induced HMGB1 release from hepatocytes by loss of autophagy Promoted ductular reaction by HMGB1 in autophagy-deficient livers Prevented ductular reaction by inhibition of HMGB1 HMGB1 and receptor for advanced glycation end product promote tumorigenesis in autophagy-deficient livers	[399]
Mouse liver (Ischemia/reperfusion; fasting)	Detections of lipidation of ATG8/LC3 and p62/SQSTM1 degradation	Protection of acute liver failure by PPARα-activated autophagy Attenuated inflammatory response by PPARα-activated autophagy	[349]
Wild type and NRBF2 knockout mice	Detections of lipidation of ATG8/LC3 and p62/SQSTM1 degradation Immunofluorescence analysis of mcherry-GFP-LC3B	Association of NRBF2 with PI(3)K complex, which is required for autophagy initiation Impaired autophagic degradation by NRBF2 knockdown Induced hepatic necrosis in NRBF2-deficient mice Prevented ER stress-mediated cytotoxicity and liver injury by NRBF2-modulated autophagy	[400]
Mouse liver (Cecal ligation and puncture)	Immunofluorescence analysis of ATG8/LC3	Limited sepsis-induced liver injury by AMPK activation Decrease cytokine induction by AMPK activation Increased liver injury by repressed AMPK-inhibited autophagy Attenuated mitochondrial dysfunction by AMPK-mediated autophagy Suppressed inflammasome and apoptosis and protection against fulminant hepatitis by AMPK-mediated autophagy	[401,402,403]
Wild type and liver specific HIF-1β knockout mice	Detection of lipidation of ATG8/LC3 Immunofluorescence analysis of GFP-LC3 Electron micrograph of autophagic vacuoles	Increased BNIP3 and BNIP3L expressions by HIF-1β in liver cells after acute ethanol treatment Limited liver injury and steatosis by deficiency of HIF-1β Protection of liver injury by FoxO3-activated autophagy in HIF-1β-knockout hepatocytes	[404]
Mouse liver (Ischemia/reperfusion)	Detections of lipidation of ATG8/LC3 and p62/SQSTM1 degradation	Amelioration of liver injury after ischemia/reperfusion by ATRA Reduced hepatic apoptosis and inflammation after ischemia/reperfusion by ATRA Promoted autophagy through Foxo3a/phosphor-Akt/Foxo1 pathway by ATRA	[405]
Mouse liver (Ischemia/reperfusion)	Detection of lipidation of ATG8/LC3 Immunofluorescence analysis of ATG8/LC3	Impaired autophagy through HO-1/calpain 2-signaling in liver steatosis Increased sensitivity of liver injury after ischemia/reperfusion by impaired autophagy	[406]
Mouse liver (Ischemia/reperfusion)	Detections of lipidation of ATG8/LC3 and p62/SQSTM1 degradation Electron micrograph of autophagic vacuoles Immunofluorescence analysis of ATG8/LC3	Protection of liver injury after ischemia/reperfusion by 1-2-cyano3-,12-dioxooleana-1,9(11)-dien-28-oyl-imidazole-mediated autophagy activation Protection of mitochondrial dysfunction and excessive ROS during liver injury after ischemia/reperfusion Reduced liver inflammation and apoptosis after ischemia/reperfusion by CDDC-Im-enhanced autophagy Involvement of Nrf2/HO-1 pathway in the autophagy-mediated protection of liver injury after ischemia/reperfusion	[407]
Mouse liver (LPS/D-GalN)	Detections of lipidation of ATG8/LC3 and p62/SQSTM1 degradation Immunofluorescence analysis of mRFP-GFP-LC3	Attenuated LPS/D-GalN- and ConA-induced acute liver failure by FK866 Amelioration of acute liver failure through FK866-induced autophagy Reduced liver injury by rapamycin-induced autophagy Activation of FK866-induced autophagy through suppressed JNK signaling Alleviated hepatotoxicity by FK866-increased autophagy	[408]
Wild type and Cd38^−/−^ mice (LPS/D-GalN)	Detections of lipidation of ATG8/LC3 and p62/SQSTM1 degradation Immunofluorescence analysis of mCherry-GFP-LC3 Electron micrograph of autophagic vacuoles	Impaired autophagy in Cd38^−/−^ hepatocytes after LPS/D-GallN-induced liver injury Protection of LPS/D-GalN-induced liver injury by CD38 Promoted autophagy by NAADP in the protection of LPS/D-GalN-induced liver injury by CD38	[409]
Wild type and liver specific KLF6 knockout mice	Detections of lipidation of ATG8/LC3 and p62/SQSTM1 degradation Electron micrograph of autophagic vacuoles Immunofluorescence analysis of mRFP-GFP-LC3	Induced KLF6 by acute human liver injury Attenuated liver regeneration and autophagy activation by KLF6 in liver after partial hepatectomy Induced autophagy by KLF6 through the binding to promoter regions of Beclin 1 and ATG7	[410]

**Table 3 ijms-20-00300-t003:** Summary of autophagy in steatosis and fatty liver diseases.

Experimental Model	Characteristics of Autophagy	Function of Autophagy	References
Wild type and liver-specific knockout of ATG7 mice Rat hepatocyte RALA255-10G Primary rat hepatocytes Human hepatoma, Huh7 and HepG2 cells	Electron micrograph of autophagic vacuoles that engulf LDs Detection of the lipidation of ATG8/LC3 Colocalization of ATG8/LC3 with LDs-associated proteins Biochemical fractionation of autophagic vacuoles that contain LDs-related proteins	Involvement of ATG8/LC3 conjugation system in the biogenesis of LDs Promotion of LDs catabolism by hepatic autophagy, i.e., “lipophagy” Degradation of lipid by hepatic autophagy Degradation of ApoB located around LDs	[243,244,310,311]
Human hepatoma cell lines, HepG2, Huh7, and Hep3B Mouse liver	Detection of the lipidation of ATG8/LC3 Immunofluorescence analysis of LC3 Electron micrograph of autophagic vacuoles that engulf LDs Immunofluorescence analysis of mRFP-GFP-LC3	Activated autophagy by thyroid hormone Enhanced autophagic flux by thyroid hormone Induced lipophagy by thyroid hormone in vitro and in vivo Induced fatty acids β-oxidation and ketosis by thyroid hormone in hepatic cells Association of the thyroid hormone-induced lipophagy to C19orf80	[346,347]
Rat liver (Carbohydrate-rich diet; hypothermic reconditioning)	Detection of the lipidation of ATG8/LC3	Limited mitochondrial defects in liver steatosis by hypothermic reconditioning Restored autophagy in liver steatosis by hypothermic reconditioning Stimulated cell apoptosis in liver steatosis by hypothermic reconditioning	[412]
Wild type and GFP-LC3-transgenic mice Isolated primary mouse hepatocytes	Detections of lipidation of ATG8/LC3 and p62/SQSTM1 degradation Electron micrograph of autophagic vacuoles that engulf mitochondria Detection of GFP-LC3-labeled punctate structure	Induction of autophagy in hepatocytes of mice livers of ethanol-induced steatosis Enhanced cell apoptosis of hepatocytes and liver injury by inhibition of autophagy Activation of mitophagy to modulate ethanol-induced liver steatosis	[345,384]
Human hepatoma cell line, HepG2 E47 cells	Detections of lipidation of ATG8/LC3 and p62/SQSTM1 degradation Immunofluorescence analysis of LC3	Induced autophagy by ethanol-treated liver cells Increased CYP2E1 activity, lipid peroxidation, and ROS formation by ethanol treatment Increased ethanol-induced LDs accumulation by interference with autophagy	[413]
Human hepatoma cell line, HepG2 and Huh7 cells Primary human hepatocytes	Electron micrograph of autophagic vacuoles Detection of the lipidation of ATG8/LC3 Immunofluorescence analysis of LC3	Inhibited autophagy by treatment of thymidine analogues Associated of autophagy to increased ROS production, lipid accumulation, and hepatic dysfunction	[414]
Primary human hepatocytes	Detection of the lipidation of ATG8/LC3 Electron micrograph of autophagic vacuoles	Reduced hepatic steatosis and enhanced survivability by exendin-4 (GLP analog) Mitigated fatty acids-induced ER stress by exendin-4 Reduced hepatic steatosis by exendin-4-induced autophagy	[415]
Human hepatoma cell line, HepG2 cells Mouse liver	Detection of GFP-LC3-labeled punctate structure Detection of the lipidation of ATG8/LC3 Electron micrograph of autophagic vacuoles	Protection of fatty acids-induced lipotoxicity and liver injury by autophagy Protection of obesity-induced liver steatosis by autophagy	[386]
Wild type and liver-specific knockout of Vps34/PI(3)KC3 mice	Detections of lipidation of ATG8/LC3 and p62/SQSTM1 degradation Electron micrograph of autophagic vacuoles Immunofluorescence analysis of mRFP-GFP-LC3	Blocked autophagic flux by knockout of Vps34/PI(3)KC3 Induced hepatomegaly and liver steatosis by knockout of Vps34/PI(3)KC3	[416]
Wild type and p73 knockout mice Human hepatoma cell line, HepG2 cells	Detections of lipidation of ATG8/LC3 and p62/SQSTM1 degradation Immunohistochemistry analysis of p62/SQSTM1	Accumulated LDs formation in liver cells by p73 deficiency Impaired autophagy in p73-deficient liver cells Repressed autophagy through downregulating ATGs gene expressions in p73-deficient liver cells	[417]
Wild type and ALCAT1 knockout mice	Detections of lipidation of ATG8/LC3 and p62/SQSTM1 degradation Electron micrograph of autophagic vacuoles that contain mitochondria Colocalization analysis of p62/SQSTM1 and LC3	Prevented the onset of high fat diet-induced NAFLD by ablation of ALCAT1 Upregulated ALCAT1 expression in the pathogenesis of NAFLD Attenuated hepatic lipogenesis and fibrosis by ALCAT1 deficiency Prevented mitochondrial dysfunction in ALCAT1-deficient liver Promoted autophagic degradation of mitochondria by deletion of ALCAT1 Induced mitochondrial fission, oxidative stress, lipid peroxidation, mtDNA depletion, and mtDNA mutation by overexpression of ALCAT1	[418]
Human hepatocyte, L02 cells	Detections of lipidation of ATG8/LC3 and p62/SQSTM1 degradation Detection of GFP-LC3-labeled punctate structure	Impaired autophagic flux by free fatty acids-induced hepatic steatosis Involvement of TFE3 in the dysfunctional hepatic lipid metabolism Enhanced autophagic flux by overexpression of TFE3 Increased autophagic degradation of LDs by overexpression of TFE3	[419]
Mouse liver	Detection of the lipidation of ATG8/LC3	Attenuated high fat diet-induced hepatic steatosis by lovastatin Induced autophagy in liver cells by statin Reduced lipid accumulation in hepatocytes of high fat diet-induced steatosis liver through PNPLA8-induced autophagy	[420]
Wild type and apobec-1 knockout mice	Electron micrograph of autophagic vacuoles that contain ER Detections of lipidation of ATG8/LC3 and p62/SQSTM1 degradation Colocalization analysis of LC3 and calnexin	Induced ER stress and autophagy by inhibition of ApoB gene expression Decreased very low density lipoprotein secretion and plasma cholesterol by inhibition of ApoB gene expression Activated autophagic degradation of ER by loss of ApoB expression Prevented hepatic steatosis by defected ApoB expression-induced ER-phagy	[421]
Human hepatoma cell line, HepG2 cells Wild type and liver-specific knockout of Rubicon	Detections of lipidation of ATG8/LC3 and p62/SQSTM1 degradation Electron micrograph of autophagic vacuoles that contain LDs	Suppressed autophagy and induced cell apoptosis in liver cells by saturated fatty acid, palmitate treatment Activated gene transcription of Rubicon and accelerated lipoapoptosis in liver cells by Palmitate Restrained autophagy and induced apoptosis by high fat diet-induced elevation of Rubicon Amelioration of high fat diet-induced liver steatosis and injury by liver specific knockout of Rubicon	[422]
Wild type and SIRT3 knockout mice	Detections of lipidation of ATG8/LC3 and p62/SQSTM1 degradation Immunofluorescence analysis of mRFP-GFP-LC3	Enhanced autophagy in liver cells of SIRT3-deficient mice Upregulation of SIRT3 in liver cells by saturated fatty acid (SFA)-rich high fat diet Aggravated liver damage and lipotoxicity induced by SFA-induced SIRT3 Increased palmitate-induced lipotoxicity by SIRT3-suppressed autophagy The suppression of autophagy by SIRT3-mediated repression of AMPK	[423]
Wild type and GFP-LC3 transgenic mice Wild type and liver specific knockout of TFE3 mice	Detections of lipidation of ATG8/LC3 and p62/SQSTM1 degradation	Impaired lysosomal and autolysosomal functions in mice livers by Gao-binge alcohol Induced alcoholic hepatitis by Gao-binge alcohol-induced impairment of TFEB Increased mTOR lysosomal translocation and activated mTORC1 in mice livers by Gao-binge alcohol Exacerbation of Gao-binge-induced liver injury in mouse liver by acute knockdown of TFEB and TFE3	[424]
Wild type and GFP-LC3 transgenic mice Isolated mouse hepatocytes	Detection of GFP-LC3-labeled punctate structure Detection of the lipidation of ATG8/LC3	Promoted autophagy in liver cells by carbamazepine treatment Alleviated acute ethanol-induced liver injury by carbamazepine treatment Reduced chronic ethanol-induced liver injury by enhanced autophagy Improved fatty liver condition in high fat diet-fed mice by enhanced autophagy	[425]
Wild type and G6Pase^−/−^ mice	Detection of the lipidation of ATG8/LC3 Immunofluorescence analysis of LC3 Immunofluorescence analysis of mRFP-GFP-LC3	Decreased autophagosome formation by loss of G6Pase Reduced autophagic flux by G6Pase deficiency Inhibited AMPK and activated mTOR signaling by loss of G6Pase Enhanced lipid accumulation and decreased β-oxidation by G6Pase deficiency-mediated repression of autophagy Reduced lipid accumulation by restored autophagy in G6Pase-knockout cells	[427]
Mouse liver	Detection of the lipidation of ATG8/LC3 Immunofluorescence analysis of LC3	Enhanced sensitivity of liver to LPS-induced steatohepatitis by alcohol Exacerbated LPS-induced ROS production and NF-kB activation by alcohol Reduced LPS-induced autophagy by alcohol Decrease ethanol and LPS-induced liver injury by rapamycin-induced autophagy	[429]
Wild type and ATG16L1^HM^ mice Primary mouse hepatocytes	Detection of the lipidation of ATG8/LC3	Induced AMPK and ULK1 activation by trehalose Enhanced autophagic flux by trehalose Reduced triglyceride accumulation by trehalose Prevented NAFLD and dyslipidemia by feeding mice trehalose	[428]
Human hepatoma, Huh7 cells	Detection of the lipidation of ATG8/LC3 Immunofluorescence analysis of LC3 Detection of GFP-LC3- and RFP-LC3-labeled punctate structure Colocalization of RFP-LC3 with LDs	Inverse correlation of autophagy level to microvesicular steatosis in HCV patients Targeted LDs to degradation by autophagy in HCV replicon cells Increased cholesterol level by impaired autophagy in HCV-infected cells	[430]
Wild type and Parkin knockout mice	Detections of lipidation of ATG8/LC3 and p62/SQSTM1 degradation Electron micrograph of autophagic vacuoles that contain LDs and mitochondria	Increased liver injury in Parkin knockout cells after alcohol treatment Enhanced hepatic steatosis and injury in Parkin knockout cells after acute-binge alcohol treatment Reduced mitophagy in Parkin knockout cells after alcohol treatment Increased lipid peroxidation in Parkin knockout cells after alcohol treatment	[431]
Wild type and Drp, Parkin, p62/SQSTM1, Nrf2, and Opa1-knockout mice		Reestablishment of mitochondrial size in hepatocytes by simultaneous loss of Drp1 and Opa1 Restored the integrity and function of liver by simultaneous loss of Drp1 and Opa1 Restored the Parkin-independent mitophagy by simultaneous loss of Drp1 and Opa1 Promoted mitochondrial ubiquitination by p62/SQSTM1-Keap1-Rbx axis Aggregated mitochondrial damage and liver injury by p62/SQSTM1-induced mitochondrial ubiquitination Rescued liver damage in NAFLD mice by Opa1 knockout-mediated turnover of mitophagy intermediate	[362]
Wild type and liver-specific knockout of ATG7 mice	Detection of the lipidation of ATG8/LC3	Induced autophagy by LPS in Kupffer cells from high fat diet-fed mice Produced TNF-α by loss of autophagy in Kupffer cells from high fat diet-fed mice Activated toll-like receptor 4 in Kupffer cells from high fat diet-fed mice	[433]
Wild type and liver-specific knockout of FIP200/RB1CC1 mice	Detections of lipidation of ATG8/LC3 and p62/SQSTM1 degradation Electron micrograph of autophagic vacuoles	Impaired autophagy in the liver of FIP200/RB1CC1 knockout mice Prevented starvation-induced triglyceride accumulation in liver cells by chronic autophagy inhibition Reduced high fat diet-induced hepatic steatosis by FIP200/RB1CC1 deficiency Induced progressive liver injury, fibrosis, and inflammation by FIP200/RB1CC1 deficiency	[432]
Wild type and ALDH2 transgenic mice Human hepatoma, HepG2 cells	Detections of lipidation of ATG8/LC3 and p62/SQSTM1 degradation	Eliminated alcohol intake-induced hepatic steatosis by ALDH2 Attenuated chronic alcohol intake-induced hepatic fat metabolic disturbance by ALDH2 Attenuated chronic alcohol-induced inflammatory response by ALDH2 Restoration of chronic alcohol-induced suppression of hepatic autophagy by ALDH2	[434]
Human hepatoma, HepG2 cells	Detections of lipidation of ATG8/LC3 and p62/SQSTM1 degradation Detection of GFP-LC3-labeled punctate structure	Induced reticulophagy in liver cells by oleic acid (OA) Aggravated lipotoxicity-induced apoptosis in OA-treated cells by inhibition of reticulophagy Decreased hepatocyte apoptosis in OA-treated cells by inhibition of mitophagy	[363]
Wild type and ob/ob mice	Detections of lipidation of ATG8/LC3 and p62/SQSTM1 degradation Detection of GFP-LC3-labeled punctate structure	Suppressed autophagic proteolysis by hepatic steatosis Interfered with acidification and clearance of autolysosome by hepatic steatosis Suppressed cathepsin activity of autolysosome by hepatic steatosis	[435]
Mouse liver Liver specimens of NAFLD patients	Detections of lipidation of ATG8/LC3 and p62/SQSTM1 degradation Electron micrograph of autophagic vacuoles Immunofluorescence analysis of LC3	Impaired hepatic autophagy in the liver cells of NASH patients Increased ER stress and accumulated p62/SQSTM1 in livers of mice fed with high fat diet and methionine choline deficient diet Induction of hepatic autophagy by short-term treatment of palmitic acid Prevented ER stress in human stellate cells treated with palmitic acid	[436]
Liver specimens of NAFLD patients	Electron micrograph of autophagic vacuoles	Accumulated autophagic vacuoles and p62/SQSTM1 aggregate in hepatocytes from NAFLD patients Suppressed cathepsin expression in hepatocytes from NAFLD patients Correlation of p62/SQSTM1 aggregate with increased hepatic inflammation in NAFLD patients	[437]

**Table 4 ijms-20-00300-t004:** Summary of autophagy in liver cancer.

Experimental Model	Characteristics of Autophagy	Function of Autophagy	References
Rat liver (Hepatectomy; DEN; amino acid deprivation)	Degradation of endogenous proteins	Less responsiveness of carcinogen-treated rat to amino acid deprivation Increased survival rate of carcinogen-treated rat for long-term deprivation of amino acid	[439]
Rat liver	Degradation of endogenous proteins Electron micrograph of autophagic vacuoles	Failed to increase protein degradation rate in tumor cells by glucagon, cyclic AMP, and nutrient deprivation No association of the enhanced proteolysis in tumor cells with any increase in the intracellular activity levels of lysosomal cathepsin enzymes Negative modulation of autophagy by growth rate of tumor cells	[440,441,442,443]
Liver specimens of HCC patients Human HCC cell lines	Electron micrograph of autophagic vacuoles Detection of GFP-LC3-labeled punctate structure	Suppressed autophagic activity in most of HCC cell lines, which correlates with the malignant phenotype of HCC Association of autophagy defect with poor diagnosis of Bcl-X_L_^+^ HCC Synergized autophagy defect and altered apoptosis in facilitation of tumor malignant differentiation	[444,445]
Huma hepatoma, HepG2 cells	Electron micrograph of autophagic vacuoles Colocalization of mitoTracker and lysoTracker	Compensatory increase of HIF-1α in HIF-2α knockdown cells Enhanced cell viability and growth in HIF-1α and HIF-2α knockdown tumor spheroids Reduced caspase-3 activity and dysregulated Bcl-X_L_ and Bax expressions in HIF-1α and HIF-2α knockdown tumor spheroids Increased autophagy in HIF-2α knockdown tumor spheroids Reversed survival advantages in HIF-2α knockdown tumor spheroids by knockdown of HIF-1α	[446]
Human hepatoma, SMMC7721 cells	Electron micrograph of autophagic vacuoles Immunofluorescence analysis of LC3	Inhibited starvation-induced autophagy by HAb18G/CD147 in hepatoma cells Enhanced cell survival of hepatoma cells by inhibited starvation-induced autophagy Downregulation of Beclin 1 expression by HAb18G/CD147	[447]
Mouse liver (DEN) Human HCC cell lines	Electron micrograph of autophagic vacuoles that contain mitochondria	Synergistic inhibition of mTOR signaling by RAD001 and BEN235 in HCC cells Synergistic inhibition of HCC progression by RAD001 and BEN235 in HCC cells Synergistic activation of autophagy by RAD001 and BEN235 in HCC cells Induced mitophagy by RAD001 and BEN235 in liver tumors	[539]
Rat liver (DEN; CQ)	Detections of lipidation of ATG8/LC3 and p62/SQSTM1 degradation Electron micrograph of autophagic vacuoles	Promoted tumor development in the dysplastic stage (Ds) of HCC by inhibition of autophagy Suppressed tumor development in the tumor formation stage (Ts) of HCC by inhibition of autophagy Promoted cell proliferation, DNA damage and inflammation in the Ds of HCC development c tumor development in the Ds of HCC by inhibition of autophagy Increased ROS production in the Ds of HCC development c tumor development in the Ds of HCC by inhibition of autophagy Reduced tumor cell proliferation and survival in the Ts of HCC by inhibition of autophagy	[448]
Wild type and liver-specific knockout of ATG5 mice	Detections of lipidation of ATG8/LC3 and p62/SQSTM1 degradation Electron micrograph of autophagic vacuoles that contain mitochondria	Induced hepatocarcinogenesis by loss of ATG5 Increased oxidative stress and DNA damage in the livers of ATG5 knockout mice Inhibited hepatocarcinogenesis in the livers of ATG5 knockout mice by N-acetylcysteine Induction of tumor suppressors by loss of ATG5	[449]
Liver specimens of HCC patients Human HCC cell lines	Detections of lipidation of ATG8/LC3 and p62/SQSTM1 degradation Immunofluorescence analysis of mRFP-GFP-LC3	Promoted autophagy by HNF1A-AS1-miR30b axis through regulating ATG5 in HCC cells Promoted tumor growth and apoptosis by HNF1A-AS1-mediated sponging has-miR-30b-5p	[450]
Human HCC cell lines Wild type and liver-specific knockout of ATG7 mice	Detections of lipidation of ATG8/LC3 and p62/SQSTM1 degradation Immunofluorescence analysis of GFP-LC3 Colocalization of GFP-LC3 and Snail	Promoted invasion of HCC cells by autophagy-induced EMT Induced autophagy-mediated EMT through TGF-β/Smad3 Rescued EMT and invasion of autophagy-deficient HCC cells by TGF-β	[451]
Wild type and liver-specific knockout of ATG7 mice	Detections of lipidation of ATG8/LC3 and p62/SQSTM1 degradation Immunofluorescence analysis of GFP-LC3 Colocalization of GFP-LC3 and Snail	Decreased levels of epithelial genes and increased mesenchymal markers levels by liver specific knockout of ATG7 Autophagic degradation of Snail through p62/SQSTM1 Impaired autophagic flux by TGF-β Inhibited TGF-β-induced EMT through autophagy-mediated Snail degradation	[452]
Liver specimens of HCC patients	Detections of lipidation of ATG8/LC3 and p62/SQSTM1 degradation Immunofluorescence analysis of GFP-LC3 Electron micrograph of autophagic vacuoles	Upregulated LC3 expression and autophagy activation in liver specimens of HCC Association of LC3 with HIF-1α and HCC recurrence Association of high expression of LC3 with tumor size and poor HCC prognosis Maintenance of intracellular ATP and activation of mitochondrial β-oxidation by autophagic removal of damaged mitochondria in HCC development	[453]
Human hepatoma, HepG2 cells	Detections of lipidation of ATG8/LC3 and p62/SQSTM1 degradation Immunofluorescence analysis of LC3 Electron micrograph of autophagic vacuoles	Induced autophagy by multi-kinase inhibitor, linifanib in HCC cells Inhibition of autophagy by linifanib through PDGFR-β signaling Potentiation of anti-tumor effects of linifanib by inhibition of autophagy in vivo and in vitro	[454]
Human and mouse hepatoma cell lines Mouse liver	Detection of lipidation of ATG8/LC3 Immunofluorescence analysis of GFP-LC3 Electron micrograph of autophagic vacuoles	Induced cell death and inhibited cell growth of hepatoma cells by ConA Induced autophagic cell death of hepatoma cells by ConA Inhibited tumor nodule formation by ConA in vivo	[455,456,457]
Liver specimens of HCC patients Human and mouse hepatoma cell lines	Detections of lipidation of ATG8/LC3 and p62/SQSTM1 degradation Immunofluorescence analysis of LC3 Electron micrograph of autophagic vacuoles	Inhibited cell growth and induced cell death of HCC cells by chemotherapy- and photodynamic therapy-induced autophagy	[383,458,459,460,461,462,463,464,465,466,467]
Human hepatoma cells Liver specimens of HCC patients Mouse liver	Detections of lipidation of ATG8/LC3 and p62/SQSTM1 degradation Immunofluorescence analysis of GFP-LC3 Electron micrograph of autophagic vacuoles	Inhibited cell growth and induced cell apoptosis by TGF-β-induced autophagy Suppressed genome instability and hepatocarcinogenesis by MAP1S-medaited autophagy activation Induced autophagic cell death in liver cancer by HDAC6 Induced autophagic cell death in liver cancer by inactivation of HDAC1 Induced cell death and growth inhibition of HCC cells by oroxylin A-induced autophagy Induced apoptosis and growth inhibition of HCC cells by MLN4924-induced autophagy Induced cell death and growth inhibition of HCC cells by ROC1 knockdown-induced autophagy Induced apoptosis and growth inhibition of HCC cells by Hedgehog inhibition-induced autophagy	[468,469,470,471,472,473,474,475]
Human hepatoma cells Liver specimens of HCC patients Mouse liver	Detections of lipidation of ATG8/LC3 and p62/SQSTM1 degradation Immunofluorescence analysis of GFP-LC3 Electron micrograph of autophagic vacuoles	Induced autophagic cell death in HCC by OSU-03012-induced autophagy Suppressed proliferation and triggered cell death in HCC cells by endostar-induced autophagy Induced cell death of HCC cells by SAHA-induced autophagy Enhanced anti-tumor effect of tetrandrine in HCC cells by autophagy Induced growth inhibition and autophagic cell death by lapatinib Inhibited HCC cell growth by ginsenoside Rh2-coordinated autophagy Involvement of autophagy in glabridin-mediated growth inhibition of HCC cells Involvement of autophagy in the eupatilin-mediated protection against oxidative stress in HCC cells Induced the pan-histone deacetylase inhibitor-induced cell death of liver cancer cells by autophagy Improved therapeutic efficacy of artocarpin by inducing autophagic cell death	[476,477,478,479,480,481,482,483,484,485]
Human hepatoma cells Liver specimens of HCC patients Mouse liver	Detections of lipidation of ATG8/LC3 and p62/SQSTM1 degradation Immunofluorescence analysis of GFP-LC3 Electron micrograph of autophagic vacuoles	Enhanced anti-tumor drugs and natural compounds-induced cell death in HCC cells by autophagy inhibition	[486,487,488,489,490,491,492,493,494]
Human hepatoblastoma cells Liver specimens of HCC patients Mouse liver	Detections of lipidation of ATG8/LC3 and p62/SQSTM1 degradation Immunofluorescence analysis of GFP-LC3 Electron micrograph of autophagic vacuoles	Increased autophagy in the liver tissues of hepatoblastoma patients Increased chemotherapy-induced cell death of hepatoblastoma cells by autophagy inhibition Promoted the growth of hepatoblastoma in vivo by autophagy	[495]
Human hepatoma cells Mouse xenograft	Detections of lipidation of ATG8/LC3 and p62/SQSTM1 degradation Immunofluorescence analysis of mRFP-GFP-LC3 Electron micrograph of autophagic vacuoles	Inhibited cell growth, colony and spheroid formation of HCC cells by bafilomycin A1 Triggered caspase-independent cell death by BAF-A1 in HCC cells Suppressed HCC growth in mouse tumor xenograft	[496]
Human hepatoma cells	Detections of lipidation of ATG8/LC3 and p62/SQSTM1 degradation Immunofluorescence analysis of GFP-LC3 Electron micrograph of autophagic vacuoles	Protection of HCC cells against hypoxia stress by activated autophagy Modulated hypoxia-induced autophagy in HCC cells by downregulated miR-375 Inhibited hypoxia-induced autophagy and autophagic flux in HCC cells by miR-375 Resensitization of HCC cells to hypoxia stress by miR-375-inhibted mitophagy	[497]
Human hepatoma cells Mouse xenograft	Detection of lipidation of ATG8/LC3 Immunofluorescence analysis of GFP-LC3 Electron micrograph of autophagic vacuoles	Enhanced proteasome inhibitors-induced apoptosis in HCC cells by autophagy inhibition Enhanced suppression of HCC growth by autophagy and proteasome inhibition	[498]
Human hepatoblastoma cells Liver specimens of HCC patients Mouse liver	Detections of lipidation of ATG8/LC3 and p62/SQSTM1 degradation Immunofluorescence analysis of GFP-LC3 Electron micrograph of autophagic vacuoles	Increased chemoresistance of HCC cells by hypoxia-induced autophagy Protection of HCC cells against DNA damage-inducing chemotherapeutic agents by autophagy Increased tolerance of oxaliplatin treatment in HCC cells by autophagy Increased cell survival of sorafenib-treated HCC cells by autophagy Involvement of autophagy in the drug resistance of cisplatin in HCC cells Protection of nilotinib-induced HCC against cellular apoptosis by autophagy Attenuated chemosensitivity of anti-tumor reagents in HCCs by lncRNA HULC-triggered autophagy	[495,496,497,498,499,500,501,502,503,504,505,506,507]
Human hepatoma cells Mouse xenograft	Detection of lipidation of ATG8/LC3 Immunofluorescence analysis of LC3 Electron micrograph of autophagic vacuoles	Inhibited ER stress by sorafenib in HCC cells Induced ER stress-induced cellular apoptosis in HCC cells by sorafenib Induced autophagosome formation by sorafenib-induced ER stress Protection of HCC cells against cell death by ER stress-induced autophagy Enhanced sorafenib-induced tumor formation by autophagy inhibition	[508]
Human hepatoma cells	Detections of lipidation of ATG8/LC3 and p62/SQSTM1 degradation Immunofluorescence analysis of GFP-LC3 Electron micrograph of autophagic vacuoles	Induced autophagy in HCC cells by low glucose and hypoxia Downregulated Bad and Bim expressions in HCC cells by autophagy Inhibited chemotherapy-induced apoptosis in HCC cells by autophagy Increase chemotherapy-induced cell death by autophagy inhibition	[510]
Wild type and liver-specific knockout of ATG7 mice Hepatocellular carcinoma cell lines	Impaired autophagy by ATG7-deficicent mice by electron microscopy analysis and detection of LC3 lipidation Degradation of p62/SQSTM1 by autophagy	Induced hepatomegaly and the swelling of hepatic cells deficient of ATG7 Accumulated peroxisomes and deformed mitochondria in the liver cells of ATG7-deficient mice Accumulated ubiquitin- and p62/SQSTM1 containing inclusion bodies in the liver cells of ATG7-deficient mice Suppression of liver dysfunction in ATG7-deficient mice by additional knockout of p62/SQSTM1 Formation of p62/SQSTM1- and Keap1-postivie inclusion bodies in ATG7-deficient hepatocytes Induction of Nrf2-dependent transcriptions of antioxidant genes in ATG7-deficient livers by p62/SQSTM1-Keap1 interaction Amelioration of liver dysfunction in ATG7-deficient livers by additional ablation of Nrf2 Exacerbation of liver injury in ATG7-deficient livers by additional knockout of Nrf2 Induction of hepatocellular carcinoma in the mice livers loss of ATG7 through activation of Nrf2	[356,377,378,379,380,381]
Wild type and mosaic knockout of ATG5 mice Wild type and liver-specific knockout of ATG7 mice	Electron micrograph of autophagic vacuoles	Spontaneous occurrence of multiple liver tumors by mosaic deletion of ATG5 Induction of oxidative stress and DNA damage response in ATG5-deficienct mouse liver Partial suppressed tumor progression in ATG7-deficient mice liver by additional ablation of p62/SQSTM1	[511]
Liver specimens of HCC patients Human hepatoma cells Wild type and liver-specific knockout of ATG7 mice	Detections of lipidation of ATG8/LC3 and p62/SQSTM1 degradation	Altered metabolic profiling in HCC cells by phosphor-p62/SQSTM1 (at serine 349) Altered metabolic profiling in autophagy-deficient mouse liver Specific accumulation of phosphor-p62/SQSTM1 (at serine 349) in HCV-positive HCC Suppressed proliferation and anti-cancer agent tolerance of HCC by inhibition of phosphorylated p62/SQSTM1-dependent Nrf2 activation	[512]
Liver specimens of HCC patients Human hepatoma cells Wild type and liver-specific knockout of p62/SQSTM1 mice	Detections of lipidation of ATG8/LC3 and p62/SQSTM1 degradation	Enhanced DEN-induced carcinogenesis in mouse liver by inducing p62/SQSTM1 Requirement of p62/SQSTM1 for the initiation of HCC in mice liver Acceleration of NASH to HCC progression by p62/SQSTM1 Induction of HCC formation by ectopic expression of p62/SQSTM1 Correlation of p62/SQSTM1 accumulation to HCC recurrence	[513,514]
Wild type, liver-specific knockout of ATG7, Nrf2 knockout, YAP knockout, ATG7/Nrf2 double knockout, ATG7/YAP double knockout mice	Detections of lipidation of ATG8/LC3 and p62/SQSTM1 degradation Immunofluorescence analysis of GFP-LC3	Maintenance of hepatic organ size and differentiation and hepatic Hippo tumor suppressor pathway by autophagy Degradation of YAP by autophagy Attenuated hepatomegaly and hepatocarcinogenesis by loss of YAP Inhibited the ATG7 deficiency-induced enlargement of liver size, fibrosis, progenitor cell expansion, hepatocarcinogenesis by additional ablation of YAP	[515]
Liver specimens of HCC patients Human hepatoma cells	Detections of lipidation of ATG8/LC3 and p62/SQSTM1 degradation Immunofluorescence analysis of GFP-LC3 Electron micrograph of autophagic vacuoles	Upregulated mitochondria fission in HCC cells Correlation of increased mitochondria fission to poor prognosis in HCC patients Promoted mitochondrial function and survival of HCC cells by mitochondrial fission in vivo and in vitro Inhibited mitochondrial apoptosis by increased mitochondrial fission-mediated autophagy	[516]
Mouse liver (DEN) Human hepatoma cells	Detection of lipidation of ATG8/LC3 Immunofluorescence analysis of GFP-LC3 Electron micrograph of autophagic vacuoles	Promoted DEN-induced hepatocarcinogenesis by adrenaline Sustained proliferation and survival of HCC cells by ADRB2 Negative modulation of autophagy by ADRB2 Inhibited autophagy by ADRB2 through interfering with PI(3)K complex formation Stabilization of HIF-1α by ADRB2-suppressed autophagic degradation Promoted HCC xenografts in vivo by ADRB2 Inhibited anti-tumor effect of sorafenib by ADRB-2-repressed autophagy	[517]
Human hepatoma cells Liver specimens of HCC patients	Detection of lipidation of ATG8/LC3 Immunofluorescence analysis of GFP-LC3	Reactivated ketolysis and facilitated cell proliferation by nutrient starvation in HCC cells Repressed AMPK activation by ketolysis in nutrient starved HCC cells Protected HCC cells from excessive autophagy and enhanced tumor growth by suppressed AMPK	[518]
Human hepatoma cells Mouse xenograft	Detection of lipidation of ATG8/LC3 Immunofluorescence analysis of LC3	Promoted autophagy by PCAF in HCC cells Induced cell death and suppressed tumor growth by PCAF-activated autophagy in vivo and in vitro	[519]
Human hepatoma cells	Detections of lipidation of ATG8/LC3 and p62/SQSTM1 degradation	Positive regulation of hepatic cancer stem cells by autophagy Requirement of p53 for autophagy to regulate hepatic cancer stem cells Phosphorylation of p53 at serine-392 by PINK1 Removal of phosphorylated p53 by mitophagy	[520]
Human cholangiocellular carcinoma cell lines	Electron micrograph of autophagic vacuoles Immunofluorescence analysis of acidic organelles	Participation in Vitamin K2-mediated growth inhibition by autophagy	[521]
Human cholangiocellular carcinoma cell lines Mouse xenograft Cholangiocellular carcinoma patient specimens	Electron micrograph of autophagic vacuoles Immunofluorescence analysis of GFP-LC3 Detection of lipidation of ATG8/LC3 Immunohistochemistry analysis of Beclin 1 Immunohistochemistry analysis of HIF-1α, BNIP3, and PI(3)KC3	Induced autophagy in cholangiocellular carcinoma cells of mice xenograft model and clinical specimens of cholangiocellular carcinoma patients Induction of apoptosis in cholangiocellular carcinoma cell lines, repression of tumor formation in cholangiocellular carcinoma xenograft mice, and enhancement of chemosensitivity of cisplatin in cholangiocellular carcinoma cells Positive correlation between low Beclin 1 expression and lymph node metastasis as well as poor survival of cholangiocellular carcinoma patients Positive correlation between high HIF-1α, BNIP3, and PI(3)KC3 expressions and lymph node metastasis as well as poor survival of cholangiocellular carcinoma patients	[523,524,525,530]
1. The Kras^G12D^ mutation and p53 deletion-induced intrahepatic cholangiocellular carcinoma (IHCC)	Immunofluorescence analysis of GFP-LC3 Detection of lipidation of ATG8/LC3 and p62/SQSTM1	Requirement of induced autophagy for cell growth in IHCC	[526,527]
Human cholangiocellular carcinoma cell lines Mouse xenograft	Immunofluorescence analysis of GFP-LC3	Participation in decitabine-induced growth suppression by autophagy	[522]
Human cholangiocarcinoma cell lines Mouse xenograft	Detection of lipidation of ATG8/LC3 Immunofluorescence analysis of acidic lysosomes	Participation in multi-drug resistance of chemotherapy by autophagy	[532]
Human cholangiocarcinoma cell lines	Electron micrograph of autophagic vacuoles Detection of lipidation of ATG8/LC3	Participation in cell survival signaling of ABC294640-treated human cholangiocarcinoma cells	[533]
Human cholangiocarcinoma cell lines Mouse xenograft	Detection of lipidation of ATG8/LC3 and p62/SQSTM1	Contribution to cell apoptosis in oblongifolin C-treated human cholangiocarcinoma cells by inhibited autophagic flux Contribution to cell apoptosis in salinomycin-repressed tumor cell growth in cholangiocarcinoma mouse xenograft model	[534,540]
Human cholangiocarcinoma cell line	Detection of lipidation of ATG8/LC3 and p62/SQSTM1	Increased sensitivity to cisplatin in human cholangiocarcinoma cells by CQ-suppressed autophagy Enhanced cell apoptosis of human cholangiocarcinoma cells by CQ-suppressed autophagy and ER stress	[528,529]
Human cholangiocarcinoma cell line	Electron micrograph of autophagic vacuoles Detection of lipidation of ATG8/LC3	Promotion on drug resistance of chemotherapy in human cholangiocarcinoma cells Protection human cholangiocarcinoma cells from compound C-induced cell apoptosis	[535,536]
Human cholangiocarcinoma cell line	Detection of lipidation of ATG8/LC3 and p62/SQSTM1 Immunofluorescence analysis of GFP-LC3	Promotion on miR-124-induced cell death in human cholangiocarcinoma cells Promotion on dihydroartemisinin-induced cell death in human cholangiocarcinoma cells through DAPK1-Beclin pathway	[537,538]

**Table 5 ijms-20-00300-t005:** Summary of autophagy in viral hepatitis.

Experimental Model	Characteristics of Autophagy	Function of Autophagy	References
Immortalized human hepatocytes (IHH) (HCV infection) Human hepatoma, Huh7.5 cells (HCV infection)	Electron micrograph of autophagic vacuoles Immunofluorescence analysis of GFP-LC3	Unknown	[541]
Human hepatoma, Huh7.5 cells (Transfection of HCV viral RNA)	Detection of lipidation of ATG8/LC3 Immunofluorescence analysis of GFP-LC3 Electron micrograph of autophagic vacuoles	Promotion on viral RNA replication	[542]
Human hepatoma, Huh7 cells (HCV infection)	Detection of lipidation of ATG8/LC3 Immunofluorescence analysis of GFP-LC3	Promotion on viral RNA replication Support on the translation of viral RNA	[548]
Human hepatoma, Huh7 cells (HCV infection)	Detection of lipidation of ATG8/LC3 Immunofluorescence analysis of GFP-LC3 Electron micrograph of autophagic vacuoles	Promotion on viral RNA replication Suppression of innate antiviral immunity	[549,591]
IHH (HCV infection) Human hepatoma, Huh7.5 cells (HCV infection)	Colocalization of autophagosomes and lysosomes	Promotion on viral RNA replication Suppression of innate antiviral immunity Protection the infected cells from cell death	[552]
Human hepatoma, Huh7 cells (HCV infection)	Detections of lipidation of ATG8/LC3 and p62/SQSTM1 degradation	Promotion on viral RNA replication Suppression of innate antiviral	[553]
Human hepatoma, Huh7 and Huh7.5.1 cells (HCV infection; transfection of HCV replicon RNA)	Detection of lipidation of ATG8/LC3	Promotion on viral RNA replication Support on the organization of replication complex for viral RNA	[544]
Human hepatoma, Huh7 cells (HCV infection)		Promotion on viral RNA replication Support on the organization of replication complex for viral RNA	[543]
Human hepatoma, Huh7.5.1 (Transfection of HCV replicon RNA)	Electron micrograph of autophagic vacuoles	Promotion on viral RNA replication Support on the organization of replication complex for viral RNA	[545]
Human hepatoma, Huh7 and Huh7.5.1 cells (HCV infection; transfection of HCV replicon RNA)	Immunofluorescence analysis of GFP-LC3	Promotion on viral RNA replication Support on the recruitment of lipid rafts for viral RNA replication	[546]
Human hepatoma, Huh7.5.1 cells (HCV infection)	Immunofluorescence analysis of GFP-LC3	Promotion on the release of viral particles	[555]
IHH (HCV infection) Human hepatoma, Huh7.5 cells (HCV infection)	Colocalization of autophagosomes and lysosomes	Promotion on the release of viral particles	[556]
Human hepatoma, Huh7 and Huh7.5.1 cells (HCV infection; transfection of HCV replicon RNA)	Immunofluorescence analysis of GFP-LC3	Promotion on the release of viral particles	[557]
Human hepatoma, Huh7 cells (Transfection of HCV replicon RNA)	Colocalization of autophagosomes and lysosomes Detection of lipidation of ATG8/LC3	Counteracting the viral-induced cell death	[558]
Human hepatoma, Huh7.5 cells (HCV infection; transfection of HCV replicon RNA)	Detection of lipidation of ATG8/LC3	Unknown	[592]
Human hepatoma, Huh7.5 cells (HCV infection)	Detection of lipidation of ATG8/LC3	Promotion on viral RNA replication	[593]
Human hepatoma, Huh7 cells (Transfection of HCV replicon RNA)	Detection of lipidation of ATG8/LC3	Unknown	[559]
Human hepatoma, Huh7 cells (HCV infection; transfection of HCV replicon RNA)	Detection of lipidation of ATG8/LC3 Detection of CMA	Promotion on HNF1α degradation	[571]
Human hepatoma, Huh7 and Huh7.5.1 cells (HCV infection; transfection of HCV replicon RNA)	Detection of lipidation of ATG8/LC3 Immunofluorescence analysis of LC3	Facilitation on virion release	[561]
Human hepatoma, Huh7.5.1 cells (HCV infection; transfection of HCV replicon RNA)	Detection of lipidation of ATG8/LC3 Electron micrograph of autophagic vacuoles	Promotion on viral RNA replication Support on the organization of replication complex for viral RNA	[571]
Human hepatoma, HepG2 cells (Transfection of HCV replicon RNA)	Detection of lipidation of ATG8/LC3 Immunofluorescence analysis of LC3	Promotion on the degradation of replicon RNA	[563,564]
Human hepatoma, Huh7 cells (HCV infection; transfection of HCV replicon RNA) Liver specimens of HCV-infected patients	Detection of lipidation of ATG8/LC3 Immunofluorescence analysis of RFP-LC3	Promotion on LDs catabolism	[430]
Human hepatoma, Huh7.5.1 cells (HCV infection; transfection of HCV replicon RNA)	Detection of lipidation of ATG8/LC3 Colocalization of autophagosome and mitochondria	Promotion on mitochondria degradation Protection of infected cells from apoptosis Establishment of viral persistence	[565,567]
Human hepatoma, Huh7 cells (HCV infection; transfection of HCV viral protein)	Detection of lipidation of ATG8/LC3 Colocalization of autophagosome and mitochondria	Sustained mitochondrial injury	[568]
Human hepatoma, Huh7 cells (HCV infection; transfection of HCV replicon RNA)	Detection of lipidation of ATG8/LC3 Colocalization of autophagosome and ubiquitinated proteins	Unknown	[571]
Human hepatoma, Huh7 and Huh7.5.1 cells (HCV infection)	Detection of lipidation of ATG8/LC3 Detection of CMA	Suppression of innate antiviral immunity	[570]
Human hepatoma, Huh7 cells (HCV infection)	Detection of lipidation of ATG8/LC3 Colocalization of autophagosome and HCV NS5A	Promotion on viral RNA replication Repression of innate antiviral immunity	[572]
Human hepatoma, Huh7.5.1 cells (HBV genomic DNA transfection) Wild type and liver specific knockout of ATG5	Detection of lipidation of ATG8/LC3 Immunofluorescence analysis of GFP-LC3 Detections of lipidation of ATG8/LC3 and p62/SQSTM1 degradation	Promotion on HBV DNA replication Induction of autophagy by HBV via HBx	[574,575,576]
Human hepatoma cell lines (HBx transfection; transfection of HBV genomic DNA)	Electron micrograph of autophagic vacuoles Immunofluorescence analysis of GFP-LC3	Transcriptional activation of Beclin 1 by HBx Activation autophagy by transfection of HBV genomic DNA	[577]
Human hepatoma, Huh7 cells (SHBs transfection; transfection of HBV genomic DNA)	Electron micrograph of autophagic vacuoles Immunofluorescence analysis of GFP-LC3 Detections of lipidation of ATG8/LC3 and p62/SQSTM1 degradation	Requirement of SHBs for autophagy activation by HBV Induction of autophagy by SHBs via ER stress Requirement of autophagy for HBV envelopment	[578]
Human hepatoma, Huh7 cells (Transfection of HBV genomic DNA)	Detection of lipidation of ATG8/LC3 Immunofluorescence analysis of GFP-LC3	Impaired HBV formation and release by silencing of ATG5, ATG12, ATG16L1 Altered intracellular distribution of HBV by silencing of ATG12 Requirement of ATG5-ATG12-ATG16L1 complex for HBV viral maturation	[579]
Human hepatoma, Huh7, HepG2, and HepG2.2.15 cells Primary human hepatocytes (Transfection of HBV genomic DNA)	Detection of lipidation of ATG8/LC3 Immunofluorescence analysis of GFP-LC3	Activated autophagy by HBV through miR-192-2p-XIAP axis Inhibited HBV replication by miR-192-2p-mediated NF-κB signaling and autophagy inhibition	[580]
Human hepatoma, HepG2.2.15 cells (Transfection of HBV genomic DNA)	Detections of lipidation of ATG8/LC3 and p62/SQSTM1 degradation	Repressed HBV replication by EGCG Suppressed HBV-induced incomplete autophagy by EGCG	[581]
Human hepatoma, HepG2.2.15 cells (Transfection of HBV genomic DNA)	Electron micrograph of autophagic vacuoles Immunofluorescence analysis of GFP-LC3 Detection of lipidation of ATG8/LC3	Colocalization of HBV components with multivesicular bodies (MVB) and autophagosomes Facilitation of HBV secretion by an activation of the endolysosomal and autophagic pathway	[582]
Human hepatic cells lines, L02 and Chang cells Human hepatoma, HepG2.2.15 cells (HBx transfection)	Detection of lipidation of ATG8/LC3 Immunofluorescence analysis of GFP-LC3	Reduced starvation-induced cell death by HBx via autophagy activation and inhibition of mitochondrial apoptotic pathway	[583]
Human hepatoma, Huh7 and HepG2 cells (Transfection of HBV genomic DNA)		Activation of ERAD by HBV replication Upregulations of ER degradation-enhancing mannosidase-like proteins (EDEMs) by HBV replication	[584]
Liver specimens of HBV-associated HCC patients Human hepatoma, Hep3B cells	Detection of lipidation of ATG8/LC3 Immunofluorescence analysis of LC3 Electron micrograph of autophagic vacuoles	Inverse correlation of autophagy level with miR-224 expression in HBV-associated HCC patients Degradation of miR-224 by autophagic pathway Suppressed tumorigenesis of HBV-associated HCC through degradation of miR-244	[585]
Human hepatoma, HepG2.2.15 cells cells (HBx transfection)	Electron micrograph of autophagic vacuoles Detection of lipidation of ATG8/LC3	Induced autophagy by HBx through PI(3)K/Akt-mTOR pathway	[586]
Human hepatic cells line, Chang cells Human hepatoma, HepG2.2.15 cells (HBx transfection)	Detection of lipidation of ATG8/LC3 Immunofluorescence analysis of GFP-LC3	Induced autophagy by HBx through activating death-associated kinase	[587]
Human hepatoma, HepG2 cells Primary human hepatocytes (HBx transfection)	Detections of lipidation of ATG8/LC3 and p62/SQSTM1 degradation	Induced autophagy by HBx through ROS-mediated JNK regulation of Beclin 1/Bcl-2 interaction	[588]
Human hepatic cells line, L02 cells Human hepatoma, Huh7, HepG2, HepG2.2.15 cells (HBx transfection)	Detections of lipidation of ATG8/LC3 and p62/SQSTM1 degradation Electron micrograph of autophagic vacuoles Immunofluorescence analysis of LC3	Induced autophagy by HBx through interacting with HMGB1	[589]
Human hepatoma, Huh7 cells (HBx transfection)	Detections of lipidation of ATG8/LC3 and p62/SQSTM1 degradation Electron micrograph of autophagic vacuoles Immunofluorescence analysis of GFP-LC3	Inhibited autophagic degradation by HBx through impairing lysosomal maturation	[590]

**Table 6 ijms-20-00300-t006:** Summary of autophagy in fibrosis and cirrhosis.

Experimental Model	Characteristics of Autophagy	Function of Autophagy	References
HTO/Z cell line Wild type and PiZ (α1-AT-deficient) mice Wild type and GFP-LC3 transgenic mice	Detection of lipidation of ATG8/LC3 Immunofluorescence analysis of GFP-LC3	Promoted degradation of α1-ATZ by autophagy-enhancing drug, CBZ Reduced α1-ATZ-induced hepatic fibrosis by CBZ	[594]
Liver specimens of fibrinogen storage disease	Electron micrograph of autophagic vacuoles	Diminished hepatocellular death in fibrinogen storage diseases by CBZ	[595]
Mouse liver (CCl_4_)	Detections of lipidation of ATG8/LC3 and p62/SQSTM1 degradation	Prevented liver fibrosis in HCC by activating MAP1S-mediated autophagy	[596]
Mouse liver (CCl_4_)	Immunofluorescence analysis of LC3	Amelioration of liver fibrosis by T-MSCs via activating autophagy	[597]
Rat liver (CCl_4_)	Detections of lipidation of ATG8/LC3 and p62/SQSTM1 degradation	Improved liver fibrosis by chloroquine through inhibiting activation of hepatic stellate cells	[598]
Human hepatic cells line, Chang cells Human liver stellate cell line, LX-2 cells	Electron micrograph of autophagic vacuoles Detection of lipidation of ATG8/LC3 Immunofluorescence analysis of LC3	Impaired autophagic flux by dihydroceramide Increased triglyceride storage in LDs and upregulated expressions of fibrosis markers by dihydroceramide Restoration of autophagic flux and reversed dihydroceramide-induced fibrogenesis by rapamycin	[599]
Rat liver (CCl_4_) LSECs from wild type and ATG7 knockout mice	Detection of lipidation of ATG8/LC3 Immunofluorescence analysis of LC3	Upregulated autophagy during LSECs capillarization in vivo and in vitro Maintenance of LSECs homeostasis by autophagy Association of loss of LSECs autophagy with insufficient antioxidant response Promoted liver fibrosis by impaired endothelial autophagy	[600]
Human liver stellate cell line, LX-2 cells Rat liver (CCl_4_) Primary human and mouse hepatic stellate cells	Detection of lipidation of ATG8/LC3 Immunofluorescence analysis of LC3 Electron micrograph of autophagic vacuoles	Involvement of HIF-1α in the regulation of autophagy-mediated activation of hepatic stellate cells Requirement of ROS-JNK1/2-depedent autophagy activation for the induction of anti-inflammation in liver fibrosis Induced fibrogenic activity in hepatic stellate cells by XBP1 through autophagy	[603]
Wild type and liver specific knockout of ATG5 mice and Nrf2 knockout mice	Detections of lipidation of ATG8/LC3 and p62/SQSTM1 degradation Electron micrograph of autophagic vacuoles	Increase apoptosis in ATG5-deficcient hepatocytes by disruption of the homeostasis of pro-and anti-apoptotic proteins Induced liver fibrosis by loss of ATG5 Suppressed ATG5 deficiency-induced liver injury, inflammation, fibrosis, and tumor development by additional ablation of Nrf2	[404]
Wild type and ATG5 knockout mice Kupffer cells isolated from Wild type and ATG5 knockout mice (CCl_4_)	Immunofluorescence analysis of LC3	Enhanced the recruitment of inflammatory cells and hepatic inflammation in ATG5-deficient mice after CCl4-induced liver fibrosis Promoted cellular apoptosis of hepatocytes in ATG5-deficeint mice after CCl_4_-inducecd liver damage Enhanced fibrogenic properties of hepatic myofibroblast in autophagy-deficient macrophages via IL1-dependent pathway	[605]
Rat liver (CCl_4_) Hepatic stellate cell, HSC-T6 cells	Immunofluorescence analysis of LC3 Electron micrograph of autophagic vacuoles Detection of lipidation of ATG8/LC3	Reduced CCl_4_-induced liver fibrosis by DMKG through inhibition of autophagy in hepatic stellate cells in vivo	[607]
Mouse liver (CCL_4_)	Electron micrograph of autophagic vacuoles Detections of lipidation of ATG8/LC3 and p62/SQSTM1 degradation	Prevented hepatic fibrosis by Quercetin through reducing hepatic stellate cell activation and inhibiting autophagy	[608]
Human liver stellate cell line, LX-2 cells Primary hepatic stellate cells	Detection of lipidation of ATG8/LC3	Amelioration of liver fibrosis by 3-MA through inhibition of autophagy and hepatic stellate cell activation	[609]
Human hepatic, Chang cells Wild type and GOLAG2 knockout mice	Detections of lipidation of ATG8/LC3 and p62/SQSTM1 degradation Electron micrograph of autophagic vacuoles Immunofluorescence analysis of LC3	Induced autophagy by knockout of GOLAG2 Promoted liver fibrosis by loss of GOLAG2 in vivo	[610]
Human liver stellate cell line, LX-2 cells Primary hepatic stellate cells	Detection of lipidation of ATG8/LC3	Enhanced hepatocyte compensatory proliferation and hepatic stellate cell growth by β-arr1 via activating autophagy Promoted liver fibrosis by β-arr1 via autophagy-mediated Snail signaling	[611]
Liver specimens of primary biliary cirrhosis (PBC) patients Mouse intrahepatic biliary epithelial cells (BECs)	Detections of lipidation of ATG8/LC3 and p62/SQSTM1 degradation Electron micrograph of autophagic vacuoles Immunofluorescence analysis of LC3	Increased expressions of p62/SQSTM1 and LC3 by various stresses in BECs Inhibited stress-induced autophagy and cellular senescence by knockdown of p62/SQSTM1 The aggregated p62/SQSTM1 in BECs in the inflamed and damaged small bile ducts in PBC	[613]
Liver specimens of primary biliary cirrhosis (PBC) patients Mouse intrahepatic biliary epithelial cells (BECs)	Immunofluorescence analysis of LC3	Increased expressions of PDC-E2 and CCO in the damaged small bile ducts in PBC Colocalization of mitochondrial antigens with LC3 in the damaged small bile ducts in PBC Contribution of deregulated autophagy to abnormal expression of mitochondrial antigens and the autoimmune pathogenesis of bile duct lesion in PBC	[614]
Liver specimens of cirrhosis patients Rat liver (CCl_4_)	Detections of lipidation of ATG8/LC3 and p62/SQSTM1 degradation Immunofluorescence analysis of LC3 Electron micrograph of autophagic vacuoles	Increased expressions of autophagy markers in human cirrhotic livers Correlation of increased autophagy markers to the degree of ductular reaction and fibrosis severity Alleviation of CCl_4_-induced liver fibrosis by autophagy inhibition	
Liver specimens of cirrhosis patients Wild type and Axin2+CD90+ transgenic rat	Detections of lipidation of ATG8/LC3 and p62/SQSTM1 degradation	Association of autophagy defect with the transition of Axin2+ cells into Axin2+CD90+ cells in liver cirrhosis to hepatocarcinogenesisv Requirement of autophagy-dependent HGF for the transition of Axin2+cells into Axin2+CD90+ cells in liver cirrhosis Prevented hepatocarcinogenesis by blockade of autophagy-dependent HGF signaling	[616]

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
