# Peer review of "Diverse Functions of Autophagy in Liver Physiology and Liver Diseases"

_ijms, 2019, doi:10.3390/ijms20020300_

Round 1
Reviewer 1 Report
The author very detailed reviewed the basic knowledge of autophagy and its functions in liver diseases. Researchers in related fields would be beneficial from reading this review paper. Thus, I recommend it to be documented in IJMS.
Author Response
Dear Reviewer:
Thank you for giving me the opportunity to resubmit our manuscript “Diverse Functions of Autophagy in Liver Physiology and Liver Diseases” to International Journal of Molecular Sciences (Manuscript ID: ijms-409394). I appreciate the thoughtful and constructive suggestions raised by the reviewers. I have improved the content of this manuscript according to the reviewers’ comments and have incorporated sections regarding the role of autophagy in the development of cholangiocarcinoma and a discussion of autophagy as an alternative pathway for cell death. The changes in the revised manuscript and a point-by-point response to each comment are listed as follows,
To Reviewer 1:
The author very detailed reviewed the basic knowledge of autophagy and its functions in liver diseases. Researchers in related fields would be beneficial from reading this review paper. Thus, I recommend it to be documented in IJMS.
Response: Thanks for reviewer’s recognition on my efforts in preparing this manuscript and recommendation on publication of this review article in IJMS.
We hope that our responses to the reviewers address all your concerns and that this version of our manuscript could meet the criteria for publication in International Journal of Molecular Sciences. Thank you for the kind considerations.

Reviewer 2 Report
In this review, Ke described the different mechanisms and types of autophagy and then focused the attention on the relevance of autophagy in chronic liver diseases as well as in hepatic neoplasias. The review is well written and very accurate, but too broad and unfocused. To better fit with the review title “Diverse Functions of Autophagy in Liver Physiology and Liver Diseases”, the first part must be reduced and limited to a brief overview on autophagy, and then the main text must be focused on the significance of autophagy on liver pathology. The first part of this review could be eventually used as “core” for another manuscript on the biology of autophagy. With regards to the chapter on liver neoplasia, a part on cholangiocarcinoma could be added.
Author Response
Dear Reviewer:
Thank you for giving me the opportunity to resubmit our manuscript “Diverse Functions of Autophagy in Liver Physiology and Liver Diseases” to International Journal of Molecular Sciences (Manuscript ID: ijms-409394). I appreciate the thoughtful and constructive suggestions raised by the reviewers. I have improved the content of this manuscript according to the reviewers’ comments and have incorporated sections regarding the role of autophagy in the development of cholangiocarcinoma and a discussion of autophagy as an alternative pathway for cell death. The changes in the revised manuscript and a point-by-point response to each comment are listed as follows,
To Reviewer 2:
In this review, Ke described the different mechanisms and types of autophagy and then focused the attention on the relevance of autophagy in chronic liver diseases as well as in hepatic neoplasias. The review is well written and very accurate, but too broad and unfocused. To better fit with the review title “Diverse Functions of Autophagy in Liver Physiology and Liver Diseases”, the first part must be reduced and limited to a brief overview on autophagy, and then the main text must be focused on the significance of autophagy on liver pathology. The first part of this review could be eventually used as “core” for another manuscript on the biology of autophagy. With regards to the chapter on liver neoplasia, a part on cholangiocarcinoma could be added.
Response: I am very grateful to reviewer’s thoughtful suggestions on section of autophagy overview. In past decades, new conclusions, ideas, and hypotheses for the regulation of autophagy and the roles of autophagy in liver diseases grew enormously. However, large discrepancies and many controversial conclusions have emerged among different studies, possibly from the different approaches and cell contexts used. Previous review articles often focused on the regulation of autophagy by liver diseases, and they almost lacked a detailed introduction to autophagy regulation in different types of stimuli and the detailed mechanism underlying how autophagy and selective autophagy are regulated. In addition, the discovery of autophagy was mostly associated with liver physiology and pathology. Therefore, I intend to provide compelling information about the history of discovery, the molecular action of autophagic processes, the regulation of autophagy and selective autophagy pathways, and new findings on autophagy research in section 2. The knowledge in section 2 will help readers to gain the compelling background of autophagy, thus allowing readers to understand how autophagy is involved in the regulation of liver physiology in section 3, and the different kinds of approaches used to analyze autophagy in liver diseases in section 4, which stage of autophagy participates in the pathogeneses of liver diseases in section 4, and the possible reasons for the disparities among different studies in section 4. It is hoped that the reviewer kindly agrees with me to keep the completeness of the autophagy overview in section 2 of the revised manuscript. Thank you again for the constructive comments.
I fully agree with the reviewer’s suggestion regarding the role of autophagy in cholangiocarcinoma development. I have extended our manuscript to briefly introduce the current knowledge about how autophagy functions in the development of cholangiocarcinoma. Please see lines 778-780 of paragraph 4 on page 35 and lines 781-816 of paragraph 1 on page 36 in the revised manuscript.
We hope that our responses to the reviewers address all your concerns and that this version of our manuscript could meet the criteria for publication in International Journal of Molecular Sciences. Thank you for the kind considerations.

Reviewer 3 Report
The review “Diverse Functions of Autophagy in Liver Physiology and Liver Diseases” by Po-Yuan Ke synthesizes the current knowledge on the functions of autophagy in hepatic metabolism and its contribution to the pathophysiology of liver diseases.
General comments
I greatly appreciated the author's effort to mention as much work as possible on the specific argument and to make this review work complete and organic on such a vast subject. In addition, I have read with interest the second paragraph in which the author makes an historical excursus of autophagy, which is not always included in the review work on this topic.
Although the work is quite complete and well set up, I believe that some changes must be made before its publication
Specific points
In the introductory part the reference to autophagy is completely missing as a form of death cellulr alternative to apoptosis. For the sake of completeness, the authors should refer to this role played by autophagy in some experimental conditions
When the authors talk about the formation of autophagic vacuole at the MAM level they should also refer to lipid rafts that has been reported to contribute to morphogenic remodeling during autolysosome formation and maturation.
Lines 164-167. The authors write: “In addition to microtubules, the histone deacetylase 6 (HDAC6)-mediated actin remodeling also promote autophagosome–lysosome fusion”. This statement is not clear and does not explain how HDAC6 can promote the fusion of autophagosomes with lysosomes. Please, explain better or remove the sentence.
Lines 173-175. The authors write: “ATG8 family proteins..........at the initial state of autophagy”. The authors should specify that this is the case of PINK1/Parkin mediated mitophagy due to starvation conditions.
Lines 765-771. When the authors talk about the role of autophagy in HCV, they should also mention its role in protecting cells from HCV-induced defects in lipid metabolism (Gastroenterology. 2012 Mar;142(3):644-653.e3).
Author Response
Dear Reviewer:
Thank you for giving me the opportunity to resubmit our manuscript “Diverse Functions of Autophagy in Liver Physiology and Liver Diseases” to International Journal of Molecular Sciences (Manuscript ID: ijms-409394). I appreciate the thoughtful and constructive suggestions raised by the reviewers. I have improved the content of this manuscript according to the reviewers’ comments and have incorporated sections regarding the role of autophagy in the development of cholangiocarcinoma and a discussion of autophagy as an alternative pathway for cell death. The changes in the revised manuscript and a point-by-point response to each comment are listed as follows,
To Reviewer 3:
Point 1: In the introductory part the reference to autophagy is completely missing as a form of death cellular alternative to apoptosis. For the sake of completeness, the authors should refer to this role played by autophagy in some experimental conditions.
Response 1: Thank you for your thoughtful suggestions. In the revised manuscript, I have added a new paragraph to introduce autophagy as an alternative pathway for cell death in section 2.4. Please see lines 329-331 of paragraph 5 on page 7 and lines 332-353 of paragraph 1 on page 8 in the revised manuscript.
Point 2: When the authors talk about the formation of autophagic vacuole at the MAM level they should also refer to lipid rafts that has been reported to contribute to morphogenic remodeling during autolysosome formation and maturation.
Response 2: Thanks for reviewer’s comments. I have extended our manuscript to summarize the findings on the functional roles of lipid rafts in the morphogenesis of autophagosome and autolysosome maturation. Please see lines 157-166 of paragraph 1 on page 4 and line 196 of paragraph 2 on page 4 to line 201 of paragraph 1 on page 5 in the revised manuscript.
Point 3: Lines 164-167. The authors write: “In addition to microtubules, the histone deacetylase 6 (HDAC6)-mediated actin remodeling also promote autophagosome–lysosome fusion”. This statement is not clear and does not explain how HDAC6 can promote the fusion of autophagosomes with lysosomes. Please, explain better or remove the sentence.
Response 3: I appreciate reviewer’s suggestions. I have revised the content on the role of HDAC6-mediated remodeling of F-actin for the formation of F-actin network and autophagosome-lysosome fusion in the quality control autophagy. Please see lines 175-179 of paragraph 2 on page 4 in the revised manuscript.
Point 4: Lines 173-175. The authors write: “ATG8 family proteins..........at the initial state of autophagy”. The authors should specify that this is the case of PINK1/Parkin mediated mitophagy due to starvation conditions.
Response 4: Thanks for reviewer’s comments. We have revised the description of the roles of ATG8 family proteins in autophagosome-lysosome fusion in PINK1/Parkin-mediated mitophagy and starvation autophagy. Please see lines 192-196 of paragraph 2 on page 4 in the revised manuscript.
Point 5: Lines 765-771. When the authors talk about the role of autophagy in HCV, they should also mention its role in protecting cells from HCV-induced defects in lipid metabolism (Gastroenterology. 2012 Mar;142(3):644-653.e3).
Response 5: Thanks for reviewer’s thoughtful suggestions. I have strengthened the study showing that HCV-activated autophagy promotes LDs catabolism to protect the infected cells from excess accumulation of lipid. Please see line 866 of paragraph 4 on page 48 to line 875 paragraph 1 on page 49 in the revised manuscript.
We hope that our responses to the reviewers address all your concerns and that this version of our manuscript could meet the criteria for publication in International Journal of Molecular Sciences. Thank you for the kind considerations.

Round 2
Reviewer 2 Report
The part on cholangiocarcinoma was added by the Author and the paragraphs added are adeqate. I have still some concerns about the lenght of the review, but overall this manuscript is extremely comprehensive and thorough.